# Proteome-wide analysis of cysteine oxidation reveals metabolic sensitivity to redox stress

Jiska van der Reest [1], Sergio Lilla [1], Liang Zheng[1,4], Sara Zanivan[1,2] & Eyal Gottlieb [1,2,3]

Reactive oxygen species (ROS) are increasingly recognised as important signalling molecules through oxidation of protein cysteine residues. Comprehensive identification of redox-regulated proteins and pathways is crucial to understand ROS-mediated events. Here, we present stable isotope cysteine labelling with iodoacetamide (SICyLIA), a mass spectrometry-based workflow to assess proteome-scale cysteine oxidation. SICyLIA does not require enrichment steps and achieves unbiased proteome-wide sensitivity. Applying SICyLIA to diverse cellular models and primary tissues provides detailed insights into thiol oxidation proteomes. Our results demonstrate that acute and chronic oxidative stress causes oxidation of distinct metabolic proteins, indicating that cysteine oxidation plays a key role in the metabolic adaptation to redox stress. Analysis of mouse kidneys identifies oxidation of proteins circulating in biofluids, through which cellular redox stress can affect whole-body physiology. Obtaining accurate peptide oxidation profiles from complex organs using SICyLIA holds promise for future analysis of patient-derived samples to study human pathologies.

[1] Cancer Research UK Beatson Institute, Switchback Road, Glasgow G61 1BD, United Kingdom. [2] Institute of Cancer Sciences, Wolfson Wohl Cancer Research Centre, University of Glasgow, Switchback Road, Glasgow G61 1QH, UK. [3] Technion Integrated Cancer Center, Faculty of Medicine, Technion - Israel Institute of Technology, 1 Efron St. Bat Galim, Haifa 3525433, Israel. [4] Present address: Pediatric Translational Medicine Institute, Shanghai Jiao Tong University School of Medicine, Shanghai 200127, China. These authors contributed equally: Jiska van der Reest, Sergio Lilla. Correspondence and requests for materials should be addressed to L.Z. (email: zhengliang@scmc.com.cn) or to S.Z. (email: s.zanivan@beatson.gla.ac.uk) or to E.G. (email: e.gottlieb@technion.ac.il)

Cancer cells are known to produce more reactive oxygen species (ROS) compared to non-transformed cells[1,2]. High levels of ROS can cause oxidation of DNA, lipids, and proteins[3]. This can have profound consequences for cellular function, as oxidation of DNA can lead to cancer via the generation of mutations and DNA strand breaks. End-products of lipid peroxidation can also act as mutagens by forming DNA adducts, whereas direct oxidation of membrane lipids can compromise membrane integrity[4]. In contrast to damaging oxidative stress, ROS have also been shown to be instrumental signalling intermediates through modification of protein cysteine residues[5]. Whereas irreversible oxidation of proteins may lead to their dysfunctionality, reversible oxidation of cysteine residues allows for modulation of activity, engagement in redox regulation, and signalling cascades[2,6,7].

To understand the full scope of ROS (as well as reactive nitrogen species), comprehensive identification of redox-regulated proteins and cellular pathways is essential. In recent years, advancements in mass spectrometry (MS) technology have brought forth several strategies to assess cysteine thiol oxidation profiles (recently reviewed in[8]). However, identifying modified cysteine residues on a whole-proteome scale remains a technical challenge, as cysteine content in proteins is low (approximately 2.3% of the proteome[9]), with an even smaller proportion that is reversibly oxidised at any given time. Another complicating factor is the broad variety of oxidative modifications that cysteine thiols can undergo, such as sulfenylation, nitrosylation, and glutathionylation, which further fractionates the cysteine proteome. Therefore, highly sensitive tools are required to detect reversibly oxidised thiols in proteomes. This constraint can be circumvented by the use of multistep protocols to enrich for cysteine-containing peptides or subpopulations of oxidised cysteine residues, such as isotope-coded affinity tags (ICAT)[10], OxICAT[11], iodoacetyl isobaric tandem mass tags (iodoTMT)[12] or OxiTMT[13], isotopic tandem orthogonal proteolysis–activity-based protein profiling (isoTOP-ABPP)[14], and other approaches using click-chemistry[15]. A disadvantage of such enrichment approaches is that they require extensive manipulation of samples, which increases the possibility of sample loss and contamination during preparation. Most importantly, specific or insufficient enrichment can introduce bias. To mitigate these issues, we developed a simple, unbiased, and robust quantitative proteomic approach (SICyLIA) to sensitively detect and accurately measure proteome-wide cysteine oxidation dynamics under conditions of acute and chronic oxidative stress. Light or heavy stable isotope-labelled iodoacetamide (IAM) is used to differentially modify free reduced cysteine thiols between two samples, and the ratio between heavy and light IAM-labelled cysteine containing peptides is used to compare the levels of reduced cysteine residues between samples. The relative change of IAM modification on a cysteine residue within a given peptide is then used as readout for cysteine thiol oxidation. This enables the identification of redox-targeted cysteine residues without discriminating for a specific oxidative modification. As reduced protein thiols are the more abundant cysteine species in cells, we achieve proteome-wide sensitivity without the need for enrichment steps. Importantly, this high sensitivity allows for the detection of smaller, but biologically relevant, changes in oxidation states. After labelling, proteomes are analysed by combining off-line high pH reversed phase chromatography for peptide fractionation, which has been shown to have exceptional peptide separation efficiency[16–18], with on-line reverse phase nano-flow ultra-high pressure liquid chromatography (UHPLC), followed by an ultra-high-field Orbitrap mass analyser[19]. Data are then analysed with the MaxQuant computational platform, which allows highly accurate peptide and protein quantification[20]. As such, the SICyLIA workflow informs on cysteine thiols at whole proteome level without the need to use enrichment steps, includes all possible cysteine thiol oxidative modifications, and is broadly accessible due to its simplicity.

In this study, SICyLIA was used to profile protein oxidation in diverse cellular models and primary tissues. Previously characterised fumarate hydratase (FH)-deficient ($Fh1^{-/-}$) mouse immortalised primary kidney epithelial cells and kidney tissues were compared to wild-type isogenic controls ($Fh1^{fl/fl}$) as a model of chronic oxidative stress[21–24]. FH is a tricarboxylic acid cycle enzyme that converts fumarate to malate, and inactivating mutations in FH are associated with the highly malignant hereditary leiomyomatosis and renal cell cancer[25]. We have previously elucidated the mechanism by which fumarate reacts with reduced glutathione (GSH) to form the covalent adduct succinicGSH in FH-deficient cells, which leads to GSH and NADPH depletion and results in chronic oxidative stress[24]. As a model of acute oxidative stress, $Fh1^{fl/fl}$ cells were treated with physiologically recoverable concentrations of hydrogen peroxide ($H_2O_2$). Our data suggest that both acute and chronic oxidative stress induce specific metabolic adaptations through oxidation of distinct metabolic proteins. Furthermore, analysis of kidney proteomes suggests that chronic intracellular oxidative stress may have profound effects on tissue remodelling and whole-body physiology through oxidation of proteins in circulating biofluids.

## Results

**Proteomic quantification of global cysteine oxidation.** Figure 1 illustrates the SICyLIA workflow to directly compare cysteine oxidation in two diverse samples on a whole proteome scale. Control and oxidatively stressed cells or tissue samples were extracted separately in the presence of either light ($^{12}C_2H_4INO$) or stable isotope-labelled heavy ($^{13}C_2D_2H_2INO$) IAM to alkylate reduced cysteine thiols (SH), coupling a carbamidomethyl (CAM) group to the cysteine residue. After labelling, equal amounts of protein extracts were mixed using a label-swap replication strategy and treated with dithiothreitol (DTT) to reduce reversibly oxidised thiols, which were subsequently blocked with n-ethylmaleimide (NEM). Proteomes were then digested and peptides fractionated using off-line high pH reversed phase chromatography prior to UHPLC-MS/MS analysis on a Q-Exactive HF. Cysteine oxidation ratios are calculated using the MaxQuant computational platform[20] based on the abundance of light and heavy CAM-modified peptide pairs for each cysteine-containing unique peptide. As IAM reacts with reduced cysteine thiols, a decrease in CAM-modification for a given peptide indicates increased cysteine oxidation. Whereas changes in the levels of reduced cysteine between samples undergoing a short-term treatment can be compared immediately (Fig. 1a), different cell lines or tissues derived from different mice have distinct proteomes and require normalisation for protein levels. For relative protein quantification, stable isotope dimethyl labelling[26] was used in conjunction with IAM labelling (Fig. 1b). This method follows a comparable workflow as described above, streamlining these parallel procedures. As shown in Fig. 1b, a fraction of the lysates used to prepare IAM-labelled samples are digested and dimethylated with either light ($H^{12}CHO/NaBH_3CN$) or heavy ($D^{13}CDO/NaBD_3CN$) formaldehyde/sodium cyanoborohydride, mixed in equal ratios using a label-swap replication strategy for independent replicates, and subjected to high pH reversed phase chromatography fractionation before UHPLC-MS/MS analysis. Protein abundance was then determined using MaxQuant. Finally, peptide oxidation ratios are normalised for protein abundance, providing the normalised oxidation ratio for each cysteine-containing peptide in the analysed proteome.

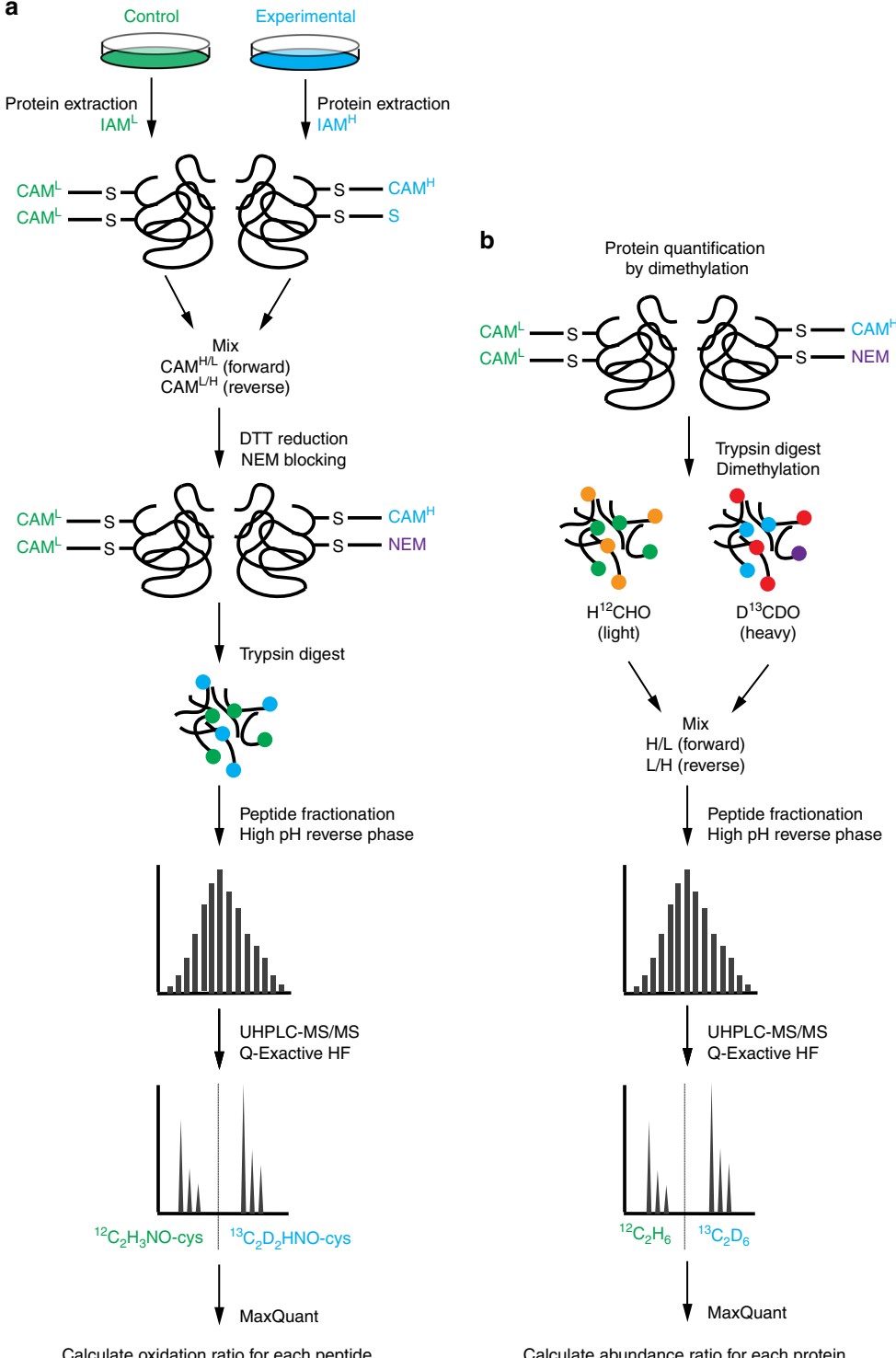

**Fig. 1** Schematic overview of the Stable Isotope Cysteine Labelling with IodoAcetamide (SICyLIA) methodology. **a** Samples are extracted in presence of either light ($^{12}C_2H_4INO$) or heavy ($^{13}C_2D_2H_2INO$) iodoacetamide (IAM) to alkylate free cysteine thiols, introducing a carbamidomethyl (CAM) group. Equal amounts of modified protein extracts are mixed, reversibly oxidised thiols are reduced with DTT and subsequently alkylated with NEM. Protein extracts are digested and peptides are fractionated prior to UHPLC-MS/MS analysis. **b** In parallel, labelled proteome extracts are trypsin digested and dimethylated using light ($H^{12}CHO/NaBH_3CN$) or heavy ($D^{13}CDO/NaBD_3CN$) formaldehyde/sodium cyanoborohydride, peptides are fractioned, and analysed using UHPLC-MS/MS similarly to IAM-modified peptides

The SICyLIA workflow was applied to diverse biological models of acute ($H_2O_2$ treatment) and chronic (*Fh1* deficiency) oxidative stress to assess its performance (Fig. 2). SICyLIA enabled quantification of 18,022 unique cysteine-containing peptides in mouse cells and 13,112 in kidney tissues (Fig. 2a). To achieve accurate cysteine oxidation ratios, stringent quality control (QC) criteria were applied. First, only peptide oxidation ratios quantified in at least three replicates were considered for

**a**

| | H$_2$O$_2$ model | *Fh1* cell model | *Fh1* tissue model |
|---|---|---|---|
| Total number of unique peptides identified | 116,175 | 132,798 | 93,980 |
| Cysteine-containing peptides identified | 18,022 | 20,848 | 13,112 |
| Peptides with oxidation ratios analysed after QC | 9479 | 8681 | 4415 |
| Corresponding protein groups | 3563 | 3006 | 2168 |
| Peptides significantly oxidised or reduced | 333 | 252 | 150 |
| Percentage significant | 3.5 | 2.9 | 3.4 |

**Fig. 2** Performance indicators and reproducibility of the SICyLIA workflow. **a** Number of peptides and proteins identified using SICyLIA in the three experimental models, broken down into indicated classes. **b** Histogram distribution and **c** boxplots of the coefficient of variation (CV%) of peptide oxidation ratios between four replicates as identified using SICyLIA for the three experimental models. Boxplots display 25th and 75th percentile (bounds of box), median (centre line), and largest and smallest value (whiskers) of the distribution. **a–c** Based on four independent experiments, single measurement (H$_2$O$_2$ model, *Fh1* cell model) or the comparison of one mouse per genotype, using four replicate tissue slices per mouse (*Fh1* tissue model)

the analysis. Additionally, the coefficient of variation (CV%) between replicates was used to filter out extreme outlier ratios (see Methods). After QC, the median oxidation ratio between replicates was used for further analysis. Oxidation ratios for up to 9479 peptides in the cells and 4415 in tissues were included for further analysis, which corresponds to 3563 and 2168 proteins, respectively. Overall, variability across our data sets was low (Fig. 2b) and replicates showed high reproducibility, with median CV% of 15.1 (H$_2$O$_2$ model), 17.7 (*Fh1* cell model), and 16.0 (*Fh1* tissue model) (Fig. 2c). To define which cysteine-containing peptides were significantly oxidised or reduced, the Significance B algorithm in the Perseus software platform was used[27]. This algorithm was specifically developed for the analysis of mass spectrometry-based protein/peptide quantification using log protein/peptide ratios[20] (see Methods). For an extended substantiation of QC strategy and discussion regarding suitability of different statistical analysis strategies for the evaluation of peptide oxidation ratios generated with SICyLIA, see Supplementary Note 1. In the analysed models, we identified 333 (H$_2$O$_2$ model), 252 (*Fh1* cell model), and 150 (*Fh1* tissue model) significantly oxidised or reduced peptides, corresponding to 3.5, 2.9, and 3.4% of the respective data sets (Fig. 2a).

**ROS induce metabolic adaptation through protein oxidation.** To provide a relevant study of acute cellular oxidative stress using hydrogen peroxide administration, a treatment regimen that induces recoverable acute oxidative stress in cells was determined. Hydrogen peroxide levels were quantified by measuring the oxidation of phenylboronate to phenol by mass spectrometry (see Methods and Supplementary Fig. 1a). This assay demonstrated sensitivity (LOD = 1 μM) and linearity in the concentration range used for this study (Supplementary Fig. 1b) and appropriate controls show stability of the detection probe over time (Supplementary Fig. 1c). Quantification of hydrogen peroxide stability in culture media showed that it decomposes slowly in the absence

of cells, but is rapidly taken up or metabolised by cells, with a half-life between 15 and 30 min (Fig. 3a). Based on these results, *Fh1$^{fl/fl}$* control cells were treated with hydrogen peroxide for 15 min, after which residual hydrogen peroxide was removed by replenishing the medium (quantification in Fig. 3a show that an average 64% of the hydrogen peroxide dose administered still remained in the medium at this point). This approach ensures that at this time point, cells are still exposed to, and detoxifying, hydrogen peroxide. Cells were considerably affected by this treatment, but retain short term proliferative capacity (Fig. 3b) as well as some long term colony formation capability (Fig. 3c and Supplementary Fig. 1d). To validate the redox stress induced by this treatment, intracellular metabolites were analysed for markers of oxidative stress. As seen in Fig. 3d, treatment with 500 μM hydrogen peroxide for 15 min led to significantly increased ratios for NAD(P)$^+$/NAD(P)H and oxidised over reduced glutathione (GSSG/GSH). Together, these results indicate that this treatment regimen caused acute and severe oxidative stress, yet cells were able to restore homoeostasis over time.

Using SICyLIA, we set out to assess protein cysteine oxidation in *Fh1$^{fl/fl}$* cells treated with hydrogen peroxide. To determine whether this 15-min treatment caused changes in global protein levels, protein abundance was compared between samples using label-free quantification (LFQ). This test was used to decide whether normalisation for protein levels (Fig. 1b) was required to compare cysteine oxidation levels between samples. Protein abundance in untreated and hydrogen peroxide-treated cells showed strong correlation (Supplementary Fig. 2a–b). Only two of the 2143 proteins quantified (0.09%) were of significantly different abundance between conditions (*t*-test analysis, FDR 0.05). Therefore, we concluded that protein normalisation was not required for this short treatment. Using the SICyLIA workflow (Fig. 1a), protein cysteine oxidation was assessed to a depth of 18,022 cysteine-containing peptides (Fig. 2a and Supplementary Data 1). Of those, 9479 were robustly quantified according to our QC criteria. These peptides belonged to 3563

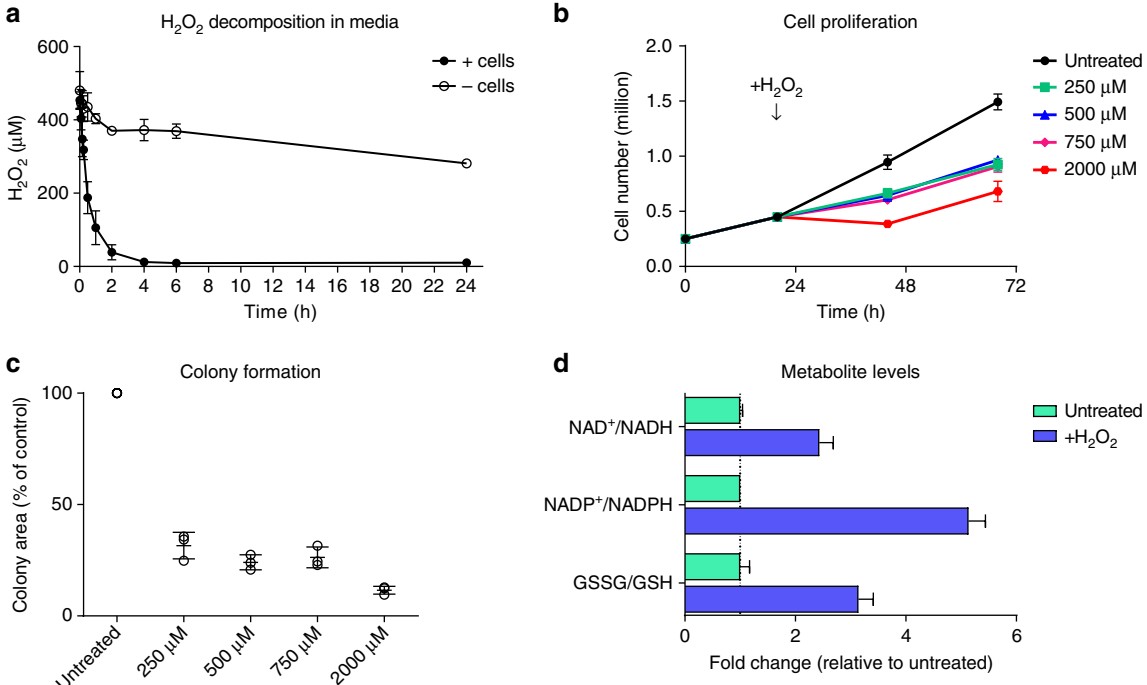

**Fig. 3** Treatment with hydrogen peroxide causes recoverable oxidative stress in *Fh1fl/fl* cells. **a** Stability of hydrogen peroxide in culture media with or without cells after treatment with 500 μM hydrogen peroxide. **b** Cell proliferation after hydrogen peroxide treatment. Media was replaced after 15 min of treatment. **c** Colony formation capacity after hydrogen peroxide treatment. Assays were started after 15 min of treatment. **d** Relative ratios of intracellular NAD+/NADH, NADP+/NADPH and GSSG/GSH after treatment with 500 μM hydrogen peroxide for 15 min. **a**–**d** All graphs show mean (S.D.) of three independent experiments with triplicate measurements

protein groups. Figure 4a displays the log2 median peptide oxidation ratio upon hydrogen peroxide treatment vs. the intensity of the peptides, with positive values (right side) representing more oxidised peptides and negative values (left side) representing more reduced peptides after treatment. As shown in Supplementary Figure 2c, the standard deviation of the replicates for the majority of peptides was low, thus justifying the use of the median oxidation ratio only for Significance B evaluation (see Supplementary Note 1) to determine significantly modified peptides. A subset of 333 peptides (3.5%) were significantly more oxidised or reduced upon treatment (Fig. 4a). To investigate the functional consequences of this global oxidation profile, we analysed whether these cysteine residues resided on functional domains (Supplementary Data 1). Uni-Prot[28] Feature Keys indicate whether cysteine residues have been shown to engage in specific functions, such as catalytic activity or metal ion binding. Figure 4b displays the Feature Keys for all cysteine residues in our dataset that were annotated in UniProt. Cysteine residues significantly modified after hydrogen peroxide treatment were found to reside predominantly in protein active sites, whereas those involved in metal ion binding were generally less reactive to oxidation upon hydrogen peroxide treatment. Cysteine residues involved in disulphide bond formation were significantly modified in specific proteins, most notably the peroxiredoxin family members (*Prdx1-6*). These are well-characterised antioxidant enzymes that control cellular peroxide levels[29]. The two-cysteine peroxiredoxin catalytic mechanism revolves around oxidation of the active site (peroxidatic) cysteine that is subsequently attacked by the resolving cysteine on the other subunit, forming a disulphide bond. In turn, these are reduced by dedicated oxidoreductase enzymes, returning the peroxiredoxins to their basal reduced state. These results indicate that the cells are actively detoxifying ROS. This was further confirmed by gene ontology (GO) category analysis[30,31] of

significantly modified peptides, which showed enrichment of proteins involved in stress and ROS response in GO biological process (GOBP; Fig. 4c) and in antioxidant, peroxiredoxin, and peroxidase activity in GO molecular function (GOMF; Fig. 4d). Additionally, a strong enrichment was found for the mitochondrial cellular compartment (GOCC; Fig. 4e).

Oxidised proteins identified using SICyLIA (Fig. 4a) included several proteins that are known to be readily oxidised. These include PTEN, PARK7, and GAPDH. Oxidation of the essential Cys124 residue of the lipid phosphatase tumour suppressor PTEN has been shown to lead to its inactivation[32]. PARK7 (also known as protein deglycase DJ-1) has been suggested to be a negative regulator of PTEN[33] as well as a scavenger of ROS[34] and stabiliser of the transcription factor NRF2, a master regulator of the cellular response to redox stress[35]. Oxidation of the Cys106 residue of PARK7 is required for its function[36]. For GAPDH, the Cys152 in its active site has been shown to be exceptionally reactive to hydrogen peroxide through a proton relay mechanism, similar to the system found in dedicated thiol peroxidases[37]. In line with the literature, our proteomics approach identified the oxidation of these particular cysteine residues (Supplementary Data 1).

Interestingly, enrichment analysis of significantly modified peptides based on GOBP showed strong enrichment for metabolic proteins (Fig. 4c). A bioinformatics comparison of cysteine content showed that metabolic proteins do not differ in cysteine content compared to other proteins (12 cysteine residues per protein on average across the whole mouse proteome, as well as in metabolic proteins defined by GOBP). The enrichment in the oxidation of cysteine residues of metabolic proteins suggests that this is a selective response to oxidative stress. To validate the functional consequences of oxidative modification, we focused on the metabolic enzyme GAPDH that had one of the largest oxidation ratios in our dataset (Fig. 4a). GAPDH catalyses the NAD+-dependent conversion of glyceraldehyde 3-phosphate

(G3P) to 1,3-bisphosphoglycerate in glycolysis. To investigate cellular metabolic alterations upon hydrogen peroxide treatment, intracellular levels of metabolites were analysed by liquid chromatography-mass spectrometry (LC-MS). Hydrogen peroxide treatment in $Fh1^{fl/fl}$ cells caused rapid accumulation of upper glycolysis intermediates up to G3P, with a concomitant decrease in downstream glycolytic metabolites, indicating a block in GAPDH activity (Fig. 5a). To investigate how GAPDH inhibition

affects the metabolic fate of glucose after oxidative stress, cells were incubated with uniformly labelled $^{13}C_6$-glucose for 15 min in the presence or absence of hydrogen peroxide and metabolites with different amount of glucose-derived $^{13}C$ isotopes (isotopologues) were analysed. The rapid accumulation of multiple isotopologues in metabolites of upper glycolysis and the pentose phosphate pathway (PPP) revealed that GAPDH inhibition by hydrogen peroxide induces rapid diversion of glycolytic flux into

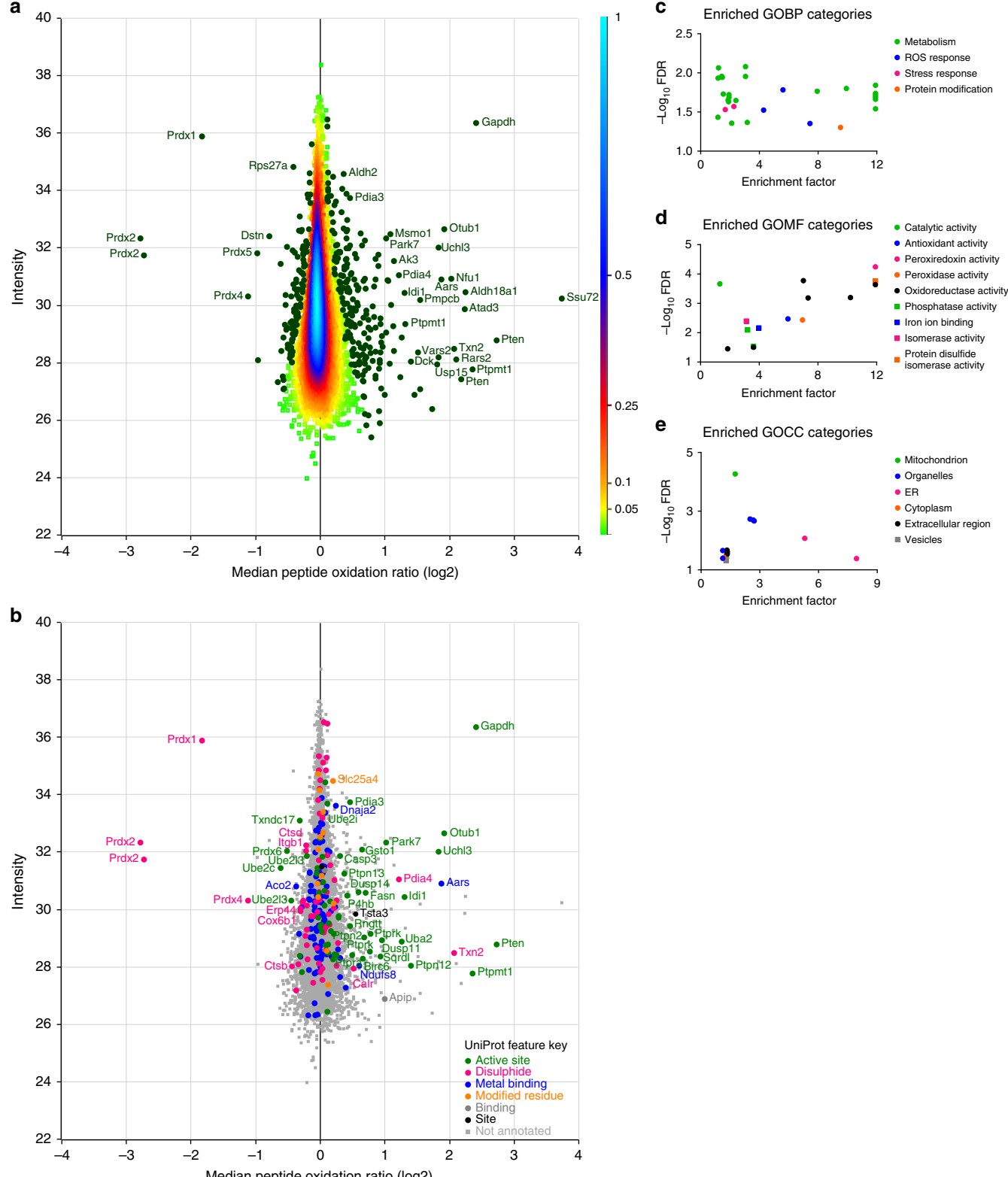

the PPP, and further, multiple rounds of cycling of carbons through the oxidative arm to amplify NADPH production and combat ROS (Fig. 5b). Furthermore, the accumulation of the GAPDH substrate G3P (Fig. 6a) and the depletion of the downstream metabolite 3-phosphoglycerate (Fig. 6b) showed a rapid return to baseline levels after the 15-min hydrogen peroxide treatment, indicating a recovery of GAPDH activity (Fig. 6c). Thus, oxidative inhibition of GAPDH allows transient partitioning of glucose carbon to the PPP for antioxidant purposes upon oxidative stress, and resumption of glucose catabolism once ROS levels fall.

In addition to the enrichment observed for metabolic proteins, enrichment analysis of significantly modified peptides showed a strong enrichment for mitochondrial proteins (Fig. 4e). It has been shown that protein cysteine content correlates with organism complexity[8]. Considering the bacterial origin of mitochondria, it is perhaps unsurprising that mitochondrial proteins were found to have a lower cysteine content than other cellular proteins in a bioinformatics comparison (8 vs. 12 cysteine residues per protein, on average). Furthermore, the steady-state redox potential of mitochondrial GSSG/GSH is more reduced as compared to the cytosol[38]. Nevertheless, it has been shown that mitochondrial proteins are highly sensitive to redox stress due to the slightly higher pH in mitochondria, which favours the reactivity of cysteine thiolates[39]. Additionally, mitochondria have a high concentration of exposed protein thiols as compared to their GSH concentration[39]. Among the significantly modified cysteine peptides after hydrogen peroxide were several components of the mitochondrial electron transport chain (ETC), such as Complex I (*Ndufa5*, *Ndufaf7*, *Ndufs8*, *Ndufv1*), Complex III (*Uqcrc2*), and Complex IV (*Cox6b1*). To investigate the consequences of this oxidation on mitochondrial function, mitochondrial oxygen consumption rate (OCR) upon treatment with hydrogen peroxide was monitored. Oxidation led to a rapid and substantial decrease in OCR (Fig. 6d). Titrating different concentrations of hydrogen peroxide showed that this was a dose-dependent effect (Supplementary Fig. 2d). Together, these results indicate that acute oxidative stress causes metabolic adaptation through the oxidation of metabolic and mitochondrial proteins to amplify the production of reducing equivalents in the PPP, and potentially to minimise the production of endogenous ROS by inhibition of mitochondrial respiration.

**Chronic redox stress causes oxidation of metabolic proteins**. To test SICyLIA on a model of higher complexity, we used the previously described immortalised primary kidney epithelial *Fh1*$^{fl/fl}$ cells from mice with a homozygous conditionally targeted *Fh1* allele[21]. *Fh1*$^{-/-}$ cells were obtained by infecting *Fh1*$^{fl/fl}$ cells with adenovirus expressing Cre recombinase[22]. These *Fh1*$^{-/-}$ cells, previously shown to experience chronic oxidative stress[24], were compared to isogenic controls (*Fh1*$^{fl/fl}$) using SICyLIA. Due to cellular adaptation to the loss of *Fh1* and concomitant changes in protein expression, this model requires normalisation for protein abundance as discussed above (Fig. 1b). Stable isotope

dimethyl labelling for relative protein quantification of *Fh1*$^{-/-}$ compared to *Fh1*$^{fl/fl}$ cells showed, unsurprisingly, that FH is the most significantly downregulated protein (Fig. 7a). Additionally, glycolytic enzymes are overexpressed (*Hk1*, *Gpi*, *Pfkl*, *Tpi1*, *Pgk1*, *Pgam1*, *Eno1*, *Pkm*, *Ldha*, *Ldhb*), as FH-deficient cells depend on glycolysis for ATP production due to truncation of the tricarboxylic acid cycle. We have previously shown that haem oxygenase (*Hmox1*) is synthetically lethal with FH[22] and as Fig. 7a indicates, HMOX1 is highly overexpressed in *Fh1*$^{-/-}$ cells. Furthermore, in line with the recent elucidation of the mechanism by which fumarate accumulation in these cells induces epithelial-to-mesenchymal-transition (EMT)[40], the protein expression data showed a clear EMT signature exemplified by downregulation of E-cadherin (*Cdh1*) and epithelial cell adhesion molecule (*Epcam*), as well as upregulation of vimentin (*Vim*) (Fig. 7a).

Similar to the depth achieved for *Fh1*$^{fl/fl}$ cells treated with hydrogen peroxide, 20,848 cysteine-containing peptides were identified (Fig. 2a). After normalisation for protein abundance, peptide oxidation ratios were obtained for 8681 peptides, which belonged to 3006 protein groups. Of those, 252 (2.9%) cysteine-containing peptides were considered significantly modified in *Fh1*$^{-/-}$ compared to *Fh1*$^{fl/fl}$ cells (Fig. 7b and Supplementary Data 2). Comparable to the acute oxidative stress model, the standard deviation for the majority of the peptides was low (Supplementary Fig. 3a). Functional annotations based on UniProt Feature Keys showed that the majority of significantly modified cysteine residues were involved in disulphide bond formation (Supplementary Fig. 3b). This was in contrast to hydrogen peroxide treatment, where the majority were part of protein active sites (Supplementary Fig. 3c). This may reflect the need for acute adaptation upon hydrogen peroxide treatment through oxidation of protein catalytic sites, such as observed for GAPDH (Figs. 5–6), whereas a different adaptive response is required for chronic oxidative stress conditions.

To further delineate cellular adaptation to redox stress, we investigated whether the preferential oxidation of metabolic proteins observed in acute oxidative stress conditions also holds true for chronic oxidative stress. This was done by enrichment analysis of the most comprehensive model of mammalian metabolism, the Recon 2 metabolic network[41]. Peptides that were considered significantly modified based on their oxidation ratio showed significant enrichment for Recon 2, in both the acute and chronic oxidative stress model (Fig. 7c). Enrichment analysis of GO categories showed that the GOBP category of cell redox homoeostasis was also significantly enriched in both models (Fig. 7c). We next investigated whether these enrichments were driven by the same proteins. Surprisingly, the majority of metabolic proteins significantly modified were mutually exclusive between the models (Fig. 7d). Of the eight proteins that were shared, only four were modified on the same cysteine residue (*Aldh18a1*, *Cox6b1*, *P4hb*, and *Pla2g4a*). Comparison of the GOBP category cell redox homoeostasis showed greater overlap between the two conditions (Fig. 7e), yet two redox proteins were

---

**Fig. 4** Acute oxidative stress causes oxidation of predominantly metabolic and mitochondrial proteins. **a** Density scatterplot displaying log 2 median peptide oxidation ratios vs. peptide intensity in *Fh1*$^{fl/fl}$ cells treated with 500 μM hydrogen peroxide for 15 min compared to untreated cells. Every square represents a unique peptide; colour scale of the density of the data points in the corresponding region is indicated on the right. Highlighted green circles are significantly oxidised (positive values) or reduced (negative values) peptides. Gene names of corresponding peptides are displayed for the most significantly oxidised or reduced peptides. **b** Scatterplot displaying log 2 median peptide oxidation ratios vs. peptide intensity in *Fh1*$^{fl/fl}$ cells treated with 500 μM hydrogen peroxide for 15 min compared to untreated cells. Every symbol represents a unique peptide, with legend indicated. Coloured circles represent peptides of which the cysteine residue has a Feature Key annotation in the UniProt database; squares represent those without this annotation. Significantly oxidised (positive values) or reduced (negative values) cysteine peptides within the annotated group are highlighted with their corresponding gene name. Enriched **c** GO biological process (GOBP), **d** molecular function (GOMF), **e** and cellular compartment (GOCC) categories within significantly oxidised or reduced proteins compared to all proteins detected. **a–e** Based on four independent experiments, single measurement

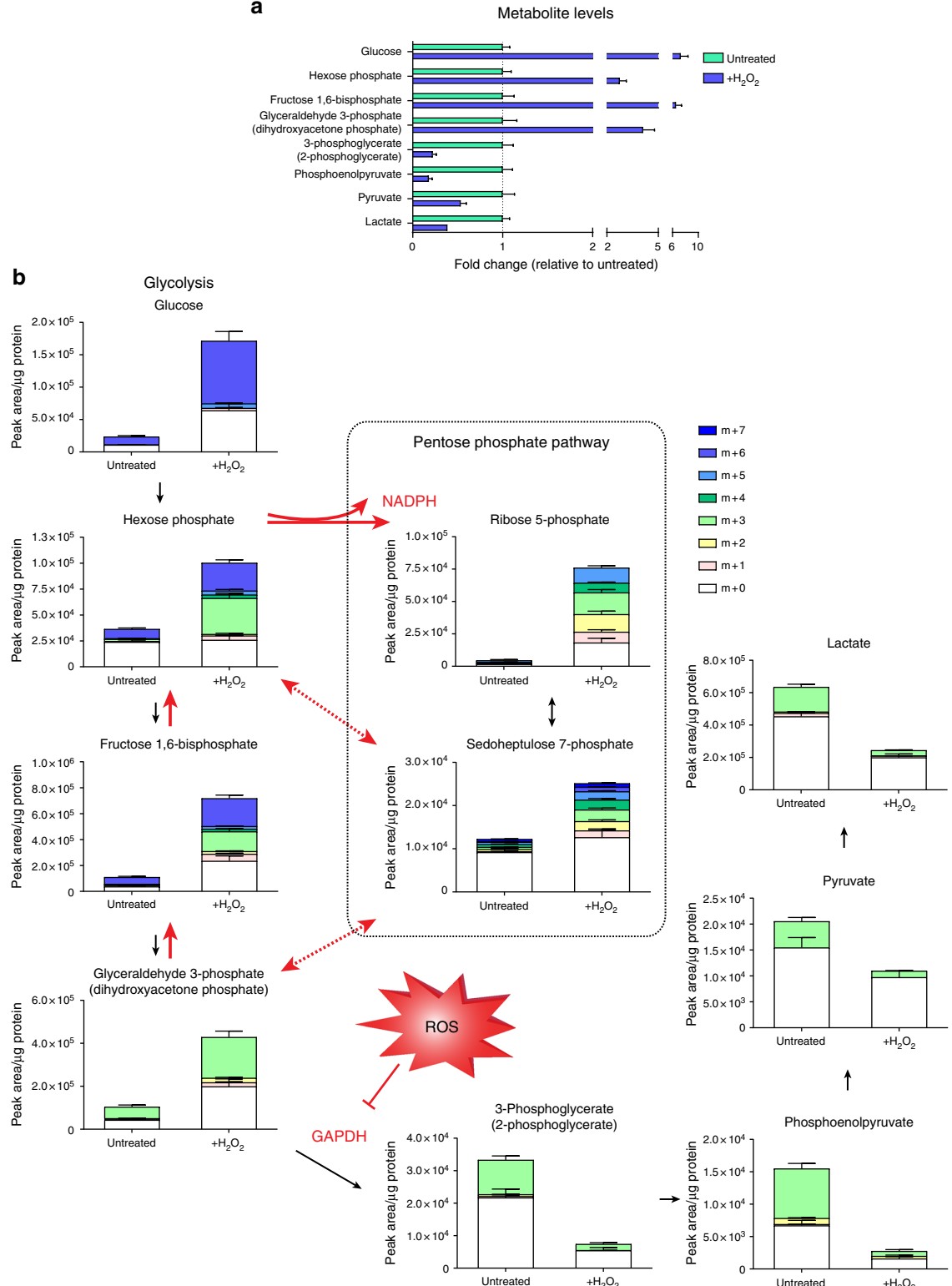

**Fig. 5** Acute oxidative stress inhibits GAPDH activity to amplify NADPH production in the PPP. **a** Relative ratios of indicated intracellular glycolytic metabolites after treatment with 500 μM hydrogen peroxide for 15 min. **b** Isotopologue distribution of intracellular metabolites in glycolysis and the pentose phosphate pathway after incubation with $^{13}C_6$-glucose and 500 μM hydrogen peroxide for 15 min. **a**, **b** All graphs show mean (S.D.) of three independent experiments with triplicate measurements

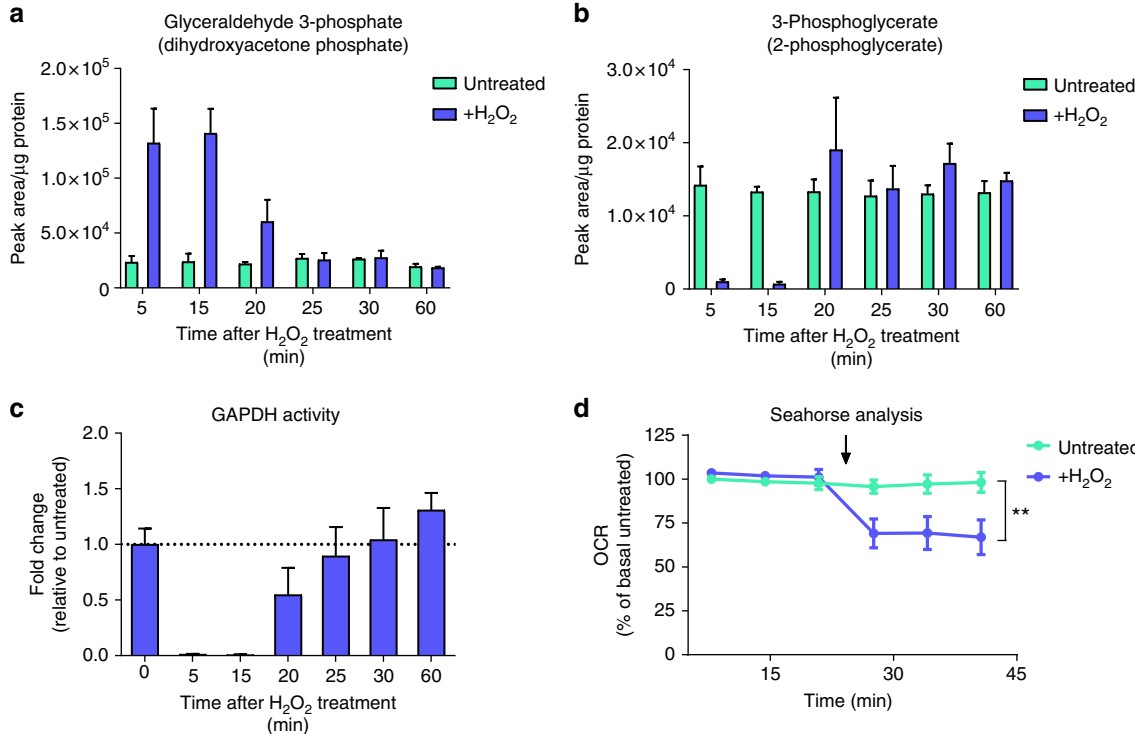

**Fig. 6** Oxidative inhibition of GAPDH is transient and acute oxidative stress reduces mitochondrial respiration. Metabolite abundance of GAPDH substrate (**a**), downstream metabolite (**b**), and resultant relative GAPDH enzymatic activity (**c**) after treatment with 500 µM hydrogen peroxide for 15 min and subsequent cellular recovery. **d** Relative oxygen consumption rate (OCR) in response to treatment with 350 µM hydrogen peroxide. Arrow indicates addition of hydrogen peroxide. **a–c** Mean (S.D.) of three independent experiments with triplicate measurements; **d** mean (S.D.) of three independent experiments with 12 replicate measurements per treatment. **$p$ value = 0.0052 (two-way ANOVA with Dunnet's test for multiple comparisons correction)

only significantly modified in chronic oxidative stress conditions, despite being detected in both models (*Gpx1* and *Txnl1*). These results indicate that acute and chronic cellular oxidative stress result in the oxidation of specific sets of proteins to help cells adapt to these distinct physiological conditions. Additionally, these results illustrate that the SICyLIA workflow is applicable to mouse-derived cells with different basal proteomes, to inform on proteins and pathways that are important in maintaining cellular homoeostasis in response to chronic oxidative stress.

**Chronic oxidative stress response in kidney tissues**. Next, we wanted to investigate whether our methodology and workflow could be applied directly to study whole organs, using the previously described *Fh1*[fl/fl] Ksp1.3/Cre genetically engineered mouse model (GEMM) of FH deficiency, which expresses Cre recombinase under the kidney-specific cadherin promoter[21]. Loss of *Fh1* in the kidney results in the formation of severe renal cysts in these mice[21], as shown in Fig. 8a. Kidney tissue slices of *Fh1*[fl/fl] Cre-negative control mice and Cre-positive mice in which *Fh1* was lost (indicated hereafter as *Fh1*[−/−]) were subjected to the same protein extraction and cysteine labelling strategies as cells, and peptide oxidation ratios were normalised for protein abundance. A total of 13,112 cysteine-containing peptides were identified (Fig. 2a). Of those, 4415 were robustly quantified peptides according to our QC criteria, and these peptides belonged to 2168 protein groups. Comparable to the other models, the standard deviation between replicates for the majority of peptides was low (Supplementary Fig. 4a). Quantification of protein cysteine oxidation in the tissues identified a subset of 150 (3.4%) significantly modified peptides (Fig. 8b and Supplementary Data 3) and functional annotations based on UniProt Feature Keys showed that nearly all significantly modified cysteine residues were

involved in disulphide bond formation (Supplementary Fig. 4b), similar to the Fh1 cell model. Additionally, similarly to the Fh1[−/−] cells, an enrichment in modified peptides of metabolic proteins (as defined by Recon 2) was also observed in the kidney tissues. Significantly modified peptides originated from various components of the tissue architecture (Fig. 8b). These included intracellular proteins, but also extracellular proteins present in the vasculature and urine. Compared to the intracellular compartment, the classical enzymatic antioxidant systems are scarce in the blood. Serum albumin (*Alb*) is the main thiol and acts as the major plasma antioxidant[42]. It was identified as the main target of oxidation in patients with various types of kidney disease (reviewed in[42]) and was oxidised on multiple peptides in the mouse kidney tissues (Fig. 8b). This was also observed for the serum protein hemopexin (*Hpx*), the scavenger protein of haemoglobin, which is produced by the liver to protect the body from free haem-induced oxidative damage[43]. A component of the Von Willebrord factor (*Vwa5a*), a blood glycoprotein involved in haemostasis whose plasma levels increase in patients with urae-mia[44], was oxidised as well. Besides plasma proteins, certain urinary proteins play a role in kidney disease and uromodulin (*Umod*), the most abundant protein excreted in urine[45] was oxidised in the kidney. Other modified proteins were extracellular matrix proteins and proteins involved in the degradation thereof (such as *Col6a1*, *Ctsb*, *Ctsd*, *Hspg2*, *Plg*, and *Vtn*) and cytoskeletal proteins (such as *Flna*, *Syne2*, *Tbcc*, and *Lpp*). Together, this may indicate that chronic oxidative stress caused by the loss of *Fh1* exerts its pathophysiological consequence of renal cyst formation in these mice through oxidation of proteins involved in metabolic processes and in maintaining tissue architecture, such as the cytoskeletal and extracellular matrix proteins, despite ROS-scavenging by intracellular, vascular, and urinary proteins.

Comparison of significantly modified proteins involved in cell redox homoeostasis showed that although some proteins were significantly modified in more than one model, others were modified in a mutually exclusive manner (Fig. 8c). Additionally, several components of the mitochondrial ETC complexes were significantly modified (Fig. 8d), especially of Complex I. With the exception of COX6B1, the components affected were distinct.

Taken together, these results further indicate that specific protein modifications are necessary for cells to adapt to distinct oxidative insults. However, four anti-oxidant proteins were modified in all three models, which may suggest that oxidative modification of these proteins is universally required in conditions of redox stress. Finally, these results indicate that similarly to the results obtained in cell lines isolated from mice, our proteomics workflow is able

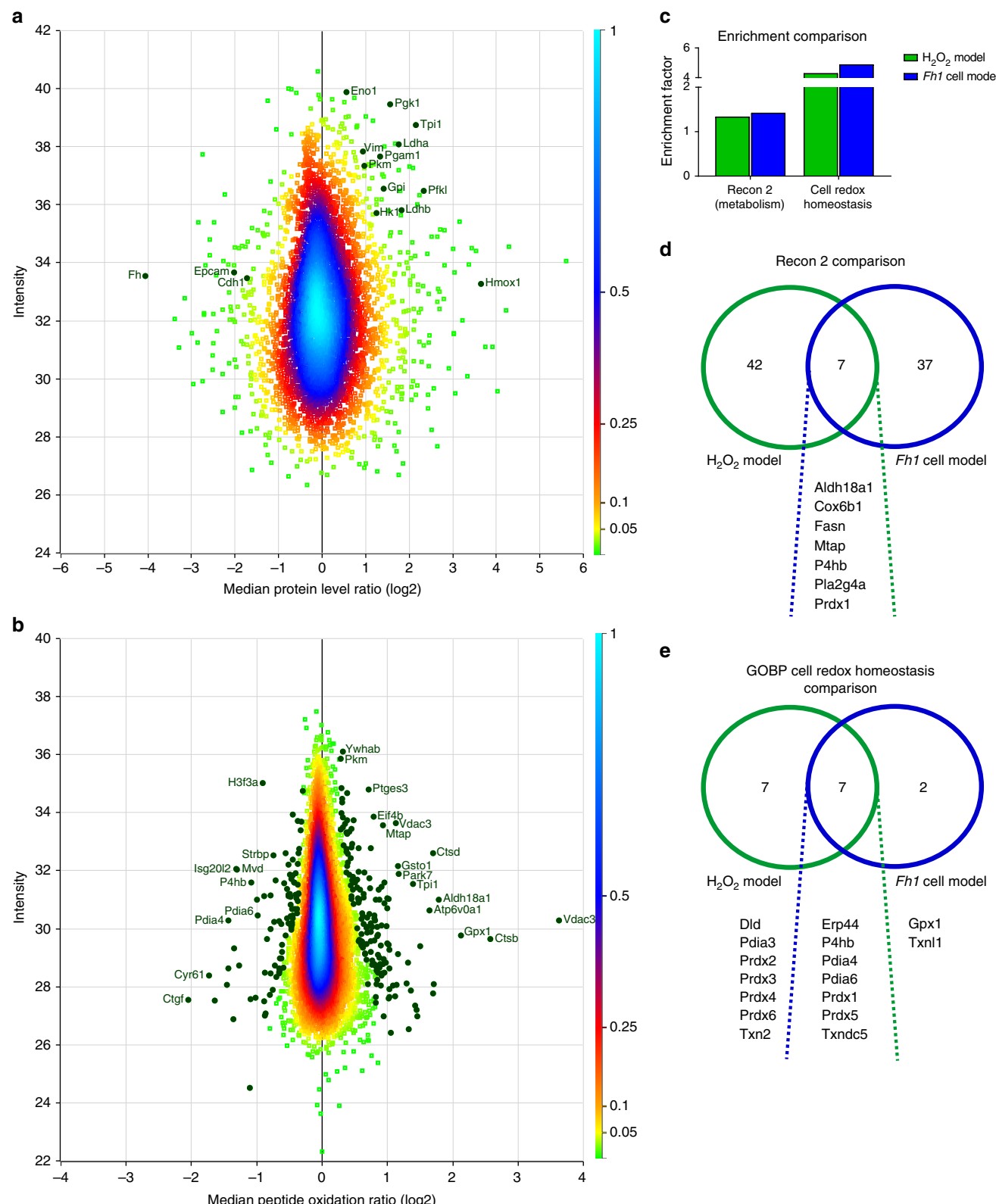

to identify proteins modified by ROS in an animal organ that contains a complex mixture of tissues and cell types. This is especially exciting given the fact that the Ksp1.3/Cre transgenic mice express Cre-recombinase solely in the epithelial cells of the renal tubules[46]. Therefore, the kidney pathology observed is caused by only a subset of cells that lose *Fh1* and experience accompanying increases in ROS[21,24]. Despite these more mild alterations in vivo as compared to the cell lines that were manipulated in vitro, the SICyLIA workflow achieved the required sensitivity to identify protein cysteine residues modified by chronic oxidative stress.

## Discussion

In recent years, attention has shifted from ROS as agents of oxidative damage to important regulators of cellular biology. While several workflows have been described to study oxidative modification of proteins, the required depth of proteomic investigation to achieve meaningful global peptide oxidation patterns remains a technical challenge. Here, we have presented SICyLIA: a comprehensive MS-based proteomic method to reliably quantify protein oxidative modification in cellular and animal models of oxidative stress. By quantifying free cysteine thiols, which are the most abundant form of cysteine in the cell, and using the loss of individual reduced thiols as readout of oxidation, we capture all possible cysteine oxidative modifications. Despite the elimination of enrichment steps, which requires extensive sample manipulation and can introduce error and bias, the SICyLIA workflow achieves a proteomic depth that enabled the identification of a large number of previously unknown redox-sensitive peptides. SICyLIA allows investigation of the whole proteome to obtain peptide oxidation and protein abundance information from the same sample, even in a complex organ with highly abundant vascular, urinary, and extracellular matrix proteins. Importantly, this is achieved using a straightforward workflow that is widely accessible to a broad audience of researchers studying oxidative stress-mediated signalling and diseases.

Recent ICAT-based studies identified ~500 ICAT-labelled cysteine-containing peptides in oxidatively stressed *S. pombe*[10] and 491 cysteine-containing peptides in *D. melanogaster*[47]. A recent IodioTMT-based approach detected 1754 labelled cysteine-containing peptides in a mammalian cell line treated with hydrogen peroxide and DTT[12]. Furthermore, the OxiTMT approach identified 1229 iodoTMT-labelled cysteine residues in *E. coli* cultures treated with hydrogen peroxide[13]. A recent study using the isoTOP-ABPP platform quantified cysteine reactivities of more than 3000 cysteine residues in a human cell line (with ~9700 cysteine residues quantified across a panel of six lines)[48]. During the preparation of this manuscript, another chemoproteomic approach titled quantitative thiol reactivity profiling (QTRP) was published, which used iodoacetamide-alkyne probes, click chemistry, and subsequent enrichment[49]. QTRP enabled quantification of cysteine reactivity to hydrogen peroxide of up to

~3700 peptides in a single human cell line and up to ~7000 across a panel of four. Our SICyLIA approach shows considerable improvement in cysteine-proteome coverage compared to all these previous methodologies, quantifying cysteine oxidation ratios for 4420 peptides in complex mouse kidney tissue samples and over 9000 in a single cell line. This is particularly notable considering the fact that SICyLIA does not require any click-chemistry or enrichment for cysteine-containing peptides.

The results obtained with SICyLIA indicate that metabolic and mitochondrial proteins are preferentially oxidised in acute oxidative stress conditions (Fig. 4a, c, e) and cysteine residues that reside on protein active sites were particularly reactive to hydrogen peroxide (Fig. 4b). Oxidation of one such active site was found on GAPDH and was part of the acute antioxidant response to inhibit GAPDH activity in favour of redirecting glucose flux into the PPP to enhance NADPH production and detoxify ROS (Figs. 5–6). Chronic oxidative stress due to *Fh1* loss caused oxidation of predominantly metabolic proteins as well (Fig. 7b, c). However, the majority of the targeted proteins were distinct (Fig. 7d) and the cysteine sites were not predominantly engaged in protein active sites (Supplementary Fig. 3c). Additionally, oxidation profiles of proteins dedicated to cell redox homoeostasis showed some exclusivity between these conditions (Fig. 7e, 8c). These results emphasise that acute oxidative stress is a distinct physiological state compared to chronic oxidative stress caused by a genetic alteration, such as loss of FH, and requires different adaptations. Taken together, our results indicate that cysteine oxidation plays a key role in the metabolic adaptation to redox stress and tailored metabolic and redox adaptations help cells respond to acute and chronic oxidative stress.

Despite these differences, there were some striking similarities between the three models investigated. Voltage-dependent anion selective channels, the most abundant protein family in the mitochondrial outer membrane, were found oxidised in all three models (*Vdac2*, *Vdac3*). Oxidation of VDAC3 in particular has been suggested to act as a sensor for mitochondrial damage, as VDAC3 over-oxidation induced removal of damaged mitochondria (recently reviewed in ref. [50]). Additionally, components of the mitochondrial ETC were modified in all models (Fig. 8d). A recent report showed that *Fh1*-deficient cells experience respiratory chain dysfunction, particularly at the level of Complex I[51]. Considering the observed drop in OCR after hydrogen peroxide treatment (Fig. 6d), ETC complex modification by ROS is likely to compromise mitochondrial respiration. As such, oxidation of VDAC2 and VDAC3 may indicate increased turnover of damaged mitochondria in response to ROS to maintain a functional mitochondrial network. Future studies using SICyLIA in a wide range of physiological models of oxidative stress will help identify whether the protein oxidative modifications observed in this study are specific to these models or present general cellular adaptations to redox stress.

In summary, our results suggest that both acute and chronic oxidative stress cause metabolic adaptation through direct oxidation

**Fig. 7** Chronic oxidative stress due to *Fh1* loss causes widespread protein oxidation. **a** Density scatterplot displaying log 2 median protein level ratios vs. protein intensity in *Fh1*$^{-/-}$ cells compared to *Fh1*$^{fl/fl}$ cells. Every square represents a unique protein; colour scale of the density of the data points in the corresponding region is indicated on the right. Positive values indicate more abundant and negative values less abundant proteins in *Fh1*$^{-/-}$ cells. Highlighted green circles are selected proteins involved in adaptation to *Fh1* loss with their corresponding gene names. **b** Density scatterplot displaying log 2 median peptide oxidation ratios vs. peptide intensity in *Fh1*$^{-/-}$ cells compared to *Fh1*$^{fl/fl}$ cells. Every square represents a unique peptide; colour scale of the density of the data points in the corresponding region is indicated on the right. Highlighted green circles are significantly oxidised (positive values) and reduced (negative values) peptides. Gene names of corresponding proteins are displayed for the most significantly oxidised or reduced peptides. **c** Comparison of enrichment analyses for the indicated categories between the acute and chronic oxidative stress model. **d** Venn diagram showing overlap and exclusivity of metabolic proteins, as defined by Recon 2, that were found significantly modified in the acute and chronic oxidative stress model. **e** Venn diagram showing overlap and exclusivity of proteins involved in cell redox homoeostasis as defined by GOBP that were found significantly modified in the acute and chronic oxidative stress model. **a–e** Based on four independent experiments, single measurement

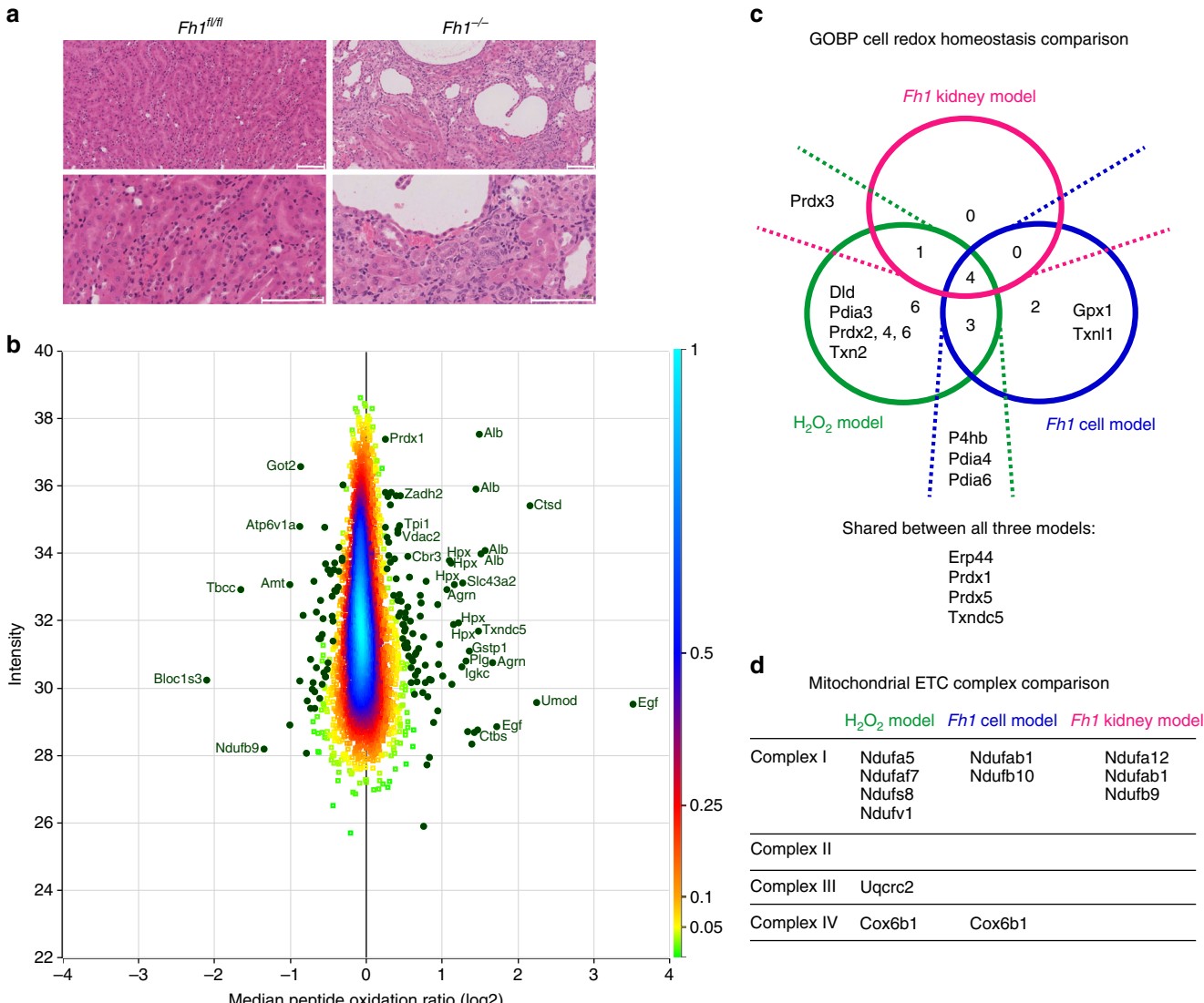

**Fig. 8** Analysis of mouse kidneys shows oxidation of proteins from various tissue compartments. **a** Representative image of $Fh1^{fl/fl}$ and $Fh1^{-/-}$ mouse kidney tissue slices stained for H&E, taken at 20× magnification. Scale bar indicates 100 μm. **b** Density scatterplot displaying log 2 median peptide oxidation ratios vs. peptide intensity in $Fh1^{-/-}$ mouse kidney tissue compared to $Fh1^{fl/fl}$ kidney tissue. Every square represents a unique peptide; colour scale of the density of the data points in the corresponding region is indicated on the right. Highlighted green circles are significantly oxidised (positive values) and reduced (negative values) peptides. Gene names of corresponding peptides are displayed for the most significantly oxidised or reduced peptides. **c** Venn diagram showing overlap and exclusivity of proteins involved in cell redox homoeostasis as defined by GOBP that were significantly modified in the acute and chronic oxidative stress cell and kidney models. **d** Comparison of components of the mitochondrial electron transport chain complexes that were significantly modified in the acute and chronic oxidative stress cell and kidney models. **a** Representative images of single histology slides; **b** based on the comparison of 1 mouse per genotype, using four replicate tissue slices per mouse. **c**, **d** Based on four independent experiments, single measurement (H₂O₂ model, $Fh1$ cell model) or the comparison of one mouse per genotype, using four replicate tissue slices per mouse ($Fh1$ tissue model)

of metabolic and mitochondrial proteins. Furthermore, chronic oxidative stress originating intracellularly can have profound effects on organ tissue architecture and physiological homoeostasis through oxidation of proteins excreted into circulating biofluids. We have demonstrated that superior sensitivity and proteomic depth can be achieved using the SICyLIA workflow. Its application to cells and tissues derived from a GEMM of FH deficiency indicates that accurate peptide oxidation profile can be obtained, even when samples are complex and basal protein levels differ between samples. This holds promise for future applications to the study of tissues derived from patients and healthy donors in studies of human diseases that are characterised by oxidative stress.

## Methods

**Sample preparation for proteomics**. For cellular proteomics, $Fh1^{fl/fl}$ cells were treated with 500 μM hydrogen peroxide 15 min prior to extraction for acute oxidation experiments. For chronic oxidation experiments, $Fh1^{fl/fl}$ and $Fh1^{-/-}$ cells were extracted without prior treatment. Plates were washed three times with phosphate-buffered saline (PBS) and extraction solution was added at 80 μl per well. Extraction solution consisted of 55 mM heavy ($^{13}C_2D_2H_2INO$) or light ($^{12}C_2H_4INO$) iodoacetamide in 100 mM Tris-HCl buffer (pH 8.5) with 4% SDS to alkylate free cysteine thiols upon lysis. Cells were scraped immediately and lysates were collected in Eppendorf tubes and sonicated (speed 2, amp 10) for 4 × 5 s. For tissue proteomics, mice were sacrificed by cervical dislocation and kidney tissues were harvested and snap frozen immediately using liquid nitrogen. Representative samples were excised on dry ice, added to extraction solution, and homogenised under dry ice vapour using a Precellys24 bead-based homogeniser (Bertin Instruments) for 3 × 20 s at 5000 rpm. All samples were centrifuged at 16,000$g$ for 5 min at 4 °C. Supernatant was transferred to new tubes and incubated in a table top shaker in the dark at 1400 rpm for 1 h at room temperature. Protein

concentration was determined by BCA assay and samples were stored at −80 °C until further processing.

For each independent experiment, 150 µg of differentially labelled protein extracts were mixed using a label-swap replication strategy (i.e. untreated-IAM-heavy mixed with hydrogen peroxide-treated-IAM-light generates forward replicate 1; untreated-IAM-light with hydrogen peroxide-treated-IAM-heavy generates reverse replicate 1, etc.), finally obtaining four replicates for each experimental model. Reversibly oxidised thiols were reduced with 70 mM DTT, incubated at room temperature (24 °C) for 45 min and diluted 1:2 using 50 mM ammonium bicarbonate solution (pH 7.0). Newly generated free thiols were subsequently alkylated using 80 mM NEM. Proteins were then precipitated in two steps using 24 and 10% solutions of trichloroacetic acid (TCA). In both steps, pellets were incubated at 4 °C for 10 min and centrifuged at 18,000 g for 5 min. Supernatants were carefully aspirated and pellets were finally washed with water until the supernatant reached neutral pH. Pellets were reconstituted in 50 µl of 8 M urea solution and submitted to a two-step digestion. First using Endoproteinase Lys-C (ratio 1:33 enzyme:lysate, Alpha Laboratories) for 1 h at room temperature (24 °C), after which partial digests were further diluted to 500 µl with 50 mM ammonium bicarbonate (pH 7.0). This reduced the urea concentration below 1 M to allow for the second digestion with trypsin (ratio 1:33 enzyme:lysate, Promega) overnight at room temperature (24 °C).

**Dimethyl labelling**. Dimethyl labelling was carried out on alkylated protein digests using a label-swap replication strategy following the on-column protocol described by Boersema et al.[26].

**Off-line HPLC fractionation**. For SICyLIA proteomic experiments, both IAM and dimethyl-modified protein digests were fractionated using reverse phase chromatography. A C18 column (150 × 2.1 mm i.d. - Kinetex EVO (5 µm, 100 Å)) was used with a Dionex HPLC system (Ultimate LPG-3000 binary pump and UVD170U Ultraviolet detector). Modules were controlled by Chromeleon version 6.7. Solvent A (98% water, 2% Acetonitrile) and solvent B (90% Acetonitrile and 10% water) were adjusted to pH 10 using ammonium hydroxide. Samples were injected manually through a Rheodyne valve onto the RP-HPLC column equilibrated with 4% solvent B and kept at this percentage for 6 min. A two-step gradient was applied at a flow-rate of 200 µl/min (from 4–27% B in 36 min, then from 27–48% B in 8 min) followed by a 5 min washing step at 80% solvent B and a 10 min re-equilibration step, for a total run time of 65 min. Column eluate was monitored at 220 and 280 nm, and collected using a Foxy Jr. FC144 fraction collector (Dionex). Collection was allowed from 9 to 54 min for 90 s per vial (300 µl) for a total of 30 fractions. No fraction concatenation strategy was used; only the first 4 and the last 5 fractions were pooled resulting in 21 fractions in total.

**UHPLC-MS/MS analysis**. For SICyLIA proteomic experiments, fractionated tryptic digests were separated by nanoscale C18 reverse-phase liquid chromatography using an EASY-nLC II 1200 (Thermo Scientific) coupled to a Q-Exactive HF mass spectrometer (Thermo Scientific). Elution was carried out using a binary gradient with buffer A (2% acetonitrile) and B (80% acetonitrile), both containing 0.1% formic acid. Samples were loaded with 8 µl of buffer A into a 20 cm fused silica emitter (New Objective) packed in-house with ReproSil-Pur C18-AQ, 1.9 µm resin (Dr Maisch GmbH). Packed emitter was kept at 35 °C by means of a column oven (Sonation) integrated into the nanoelectrospray ion source (Thermo Scientific). Peptides were eluted at a flow rate of 300 nl/min using different gradients optimised for three sets of fractions: 1–7, 8–15, and 16–21. Two-step gradients were used, all with 20 min for step one and 7 min for step two. Percentages of buffer B (%B) were changed as follows. For F1-7, %B was 2 at the start, 20 at step one, and 39 at step two. For F8-14, %B was 4 at the start, 23 at step one, and 43 at step two. For F15-21, %B was 6 at the start, 28 at step one, and 48 at step two.

All gradients were followed by a washing step (100% B) for 10 min followed by a 5 min re-equilibration step (5%), for a total run time of 40 min. Eluting peptides were electrosprayed into the mass spectrometer using a nanoelectrospray ion source (Thermo Scientific). An Active Background Ion Reduction Device was used to decrease air contaminants signal level.

For label-free quantitation analysis, proteins were digested as described above and resulting tryptic peptides were desalted with a C-18 Stage-Tip[52] and subsequently injected using an EASY-nLC 1200 (Thermo Fisher Scientific) coupled online to an Orbitrap Q-Exactive HF mass spectrometer. Peptides were eluted at 300 nl/min flow into a 50 cm fused silica emitter (New Objective) packed in-house with ReproSil-Pur C18-AQ, 1.9 µm resin (Dr Maisch GmbH). The gradient used started at 2% of buffer B and was increased to 16% over 185 min, and then to 28% over 30 min. Finally, a washing step at 95% of B was carried out over 10 min followed by a 13 min re-equilibration at 5% B for a total duration of 243 min.

Ionisation conditions used include: spray voltage 2.1 kV, ion transfer tube temperature 250 °C. Data were acquired using Xcalibur software (Thermo Scientific) and acquisition was carried out in positive ion mode using data dependent acquisition. A full scan (FT-MS) over mass range of 375–1400 m/z was acquired at 60,000 resolution at 200 m/z, with a target value of 3,000,000 ions for a maximum injection time of 20 ms. Higher energy collisional dissociation fragmentation was performed on the 15 most intense ions, for a maximum

injection time of 50 ms, or a target value of 50,000 ions. Multiply charged ions having intensity greater than 12,000 counts were selected through a 1.5 m/z window and fragmented using normalised collision energy of 27. Former target ions selected for MS/MS were dynamically excluded for 25 s.

**Proteomics data analysis**. The MS Raw data were processed with MaxQuant software[20] version 1.5.5.1 and searched with Andromeda search engine[53], querying UniProt[28] Mus musculus (20/06/2016; 57,258 entries). First and main searches were performed with precursor mass tolerances of 20 ppm and 4.5 ppm, respectively, and MS/MS tolerance of 20 ppm. The minimum peptide length was set to six amino acids and specificity for trypsin cleavage was required, allowing up to two missed cleavage sites. Methionine oxidation and N-terminal acetylation were specified as variable modifications, no fixed modifications were specified. The peptide, protein, and site false discovery rate (FDR) was set to 1 %. Modification by light and heavy iodoacetamide on cysteine residues (carbamidomethylation) was set as label type modification in Andromeda configuration. Compositions set in the software were: HNOCx(2)Hx(2) for heavy and H(3)NOC(2) for light label. As such, cysteine-containing peptide pairs were treated in the same way as SILAC pairs, and highly accurate median peptide ratios were obtained. These ratios were then further normalised to the median of all ratios in each replicate. As the MaxQuant algorithm searches for and identifies only peptides that contain either light or heavy carbamidomethylation on cysteine residues, peptides with NEM labelling were not considered. Therefore, peptides with multiple cysteine residues that carried both IAM and NEM modifications were excluded from the analysis. In our data sets, 15–16% of peptides contain multiple cysteine residues and less than 3% (H₂O₂ model), 2% (Fh1 cell model), and 4% (Fh1 tissue model) carried mixed labelling and were excluded.

For Fh1 cell and kidney tissue samples that required protein expression normalisation, dimethylated samples were processed using: DimethLys0/Nter0 and DimethLys8/Nter8 as light and heavy labels, respectively. Protein abundance was then determined by MaxQuant, which calculates the median of the ratios between light and heavy dimethyl modifications measured for all unique peptides from each protein. Both data sets (iodoacetamide heavy/light and dimethyl heavy/light) were processed at the same time in MaxQuant using different parameters, which were defined with the Parameter Groups option. Quantitation of cysteine oxidation reported in the MaxQuant output peptide.txt file was used for the analysis. For LFQ of proteins in hydrogen peroxide-treated and untreated samples, proteins were quantified according to the LFQ algorithm in MaxQuant[54].

MaxQuant output was further processed and analysed using Perseus software version 1.5.5.3[27]. Peptides with Cys count lower than one were excluded, together with Reverse and Potential Contaminant flagged peptides. Protein level quantitation was done using the ProteinGroups.txt file. From the ProteinGroups.txt file, Reverse and Potential Contaminant flagged proteins were removed, and at least one uniquely assigned peptide and a minimum ratio count of 2 were required for a protein to be quantified. Only cysteine-containing peptides uniquely assigned to one protein group within each replicate experiment were normalised and included in the analysis. Only cysteine-containing peptides and protein groups that were robustly quantified in three out of four replicate experiments (cells) or replicate samples (tissues) were used for the analysis. QC was imposed by detecting and excluding outliers by calculating upper fences (Q3 + 1.5IQR, where Q3 is the third quartile and IQR is the interquartile range) of the distribution of CV% for all median peptide oxidation ratios in each dataset and using these as cut-off. This led to the exclusion of 578 (H₂O₂ model), 499 (Fh1 cell model), and 315 (Fh1 tissue model) peptides. Throughout all further analyses, median peptide and protein ratios were used to further minimise the effect of outliers.

**Cell culture**. Fh1^{fl/fl} and Fh1^{−/−} mouse primary kidney epithelial cells were isolated, immortalised, and authenticated in our laboratory as described previously[22]. Briefly, kidney epithelial cells from mice with a homozygous conditionally targeted Fh1 allele were immortalised (referred to as Fh1^{fl/fl} cells; these express Fh1 protein at a normal level). Next, Fh1^{−/−} cells were obtained by infecting Fh1^{fl/fl} cells with adenovirus expressing Cre recombinase. Cells were maintained in DMEM (Invitrogen 21969-035, Thermo Fisher Scientific) supplemented with 2 mM glutamine and 10% FBS. All experiments were carried out in medium containing physiological concentrations of nutrients based on a formulation previously reported by our lab[55], containing 5.56 mM glucose, 0.65 mM glutamine, 0.1 mM sodium pyruvate, and 2.5% FBS (referred to as experimental medium). Cells were plated in six-well plates (Fh1^{fl/fl} at 2.5 × 10⁵ cells per well, Fh1^{−/−} at 3.75 × 10⁵ cells per well) using 2 ml medium per well and experiments were started after incubating for 20 h at 37 °C and 5% CO₂ for all experiments, except Seahorse analysis (specified below). Cell lines were negative for mycoplasma as determined by MycoAlert™ Mycoplasma Detection Kit (Lonza).

**Mouse studies**. For tissue proteomics, previously described male Fh1^{fl/fl} Ksp1.3/Cre mice were used that express Cre recombinase under the kidney-specific cadherin (Ksp-cadherin) promoter[21]. One Cre-positive mouse (referred to as Fh1^{−/−}, 232 days old) was compared to a Cre-negative control mouse (referred to as Fh1^{fl/fl}, 239 days old). Mice were sacrificed by cervical dislocation and kidneys were harvested and snap frozen immediately. Four representative kidney tissue slices were used per mice, two from each kidney. No statistical method was used to

predetermine sample size. The experiments were not randomised. The investigators were not blinded to allocation during experiments and outcome assessment. Animal work was carried out with ethical approval from the University of Glasgow under the Animal (Scientific Procedures) Act 1986 and the EU Directive 2010 (PPL 70/8645).

**Hydrogen peroxide quantification**. Phenylboronate ($C_6H_7BO_2$, Sigma 78181) was used as a probe to detect and quantify hydrogen peroxide, developed based on the methodology described in[56]. In alkaline conditions, the nucleophilic peroxide attacks the boronic acid and forms a negatively charged tetrahedral boronate intermediate. The weak C–B bond undergoes a 1,2-insertion, which makes the C-bond migrate to the peroxide oxygen atoms. The newly formed borate ester is quickly hydrolysed by water to phenol. Briefly, reaction solution was prepared by diluting phenylboronate (stock solution 60 mM in EtOH) to 1 mM using TBS-T (pH 11 for experimental samples and positive control, pH 7.5 for negative control). To detect hydrogen peroxide in media, 30 µl of medium was added to 200 µl of reaction solution. As a positive control, 30 µl of 500 µM hydrogen peroxide in water was added to 200 µl of reaction solution. Additionally a standard curve was generated for quantification purposes using various concentrations of hydrogen peroxide. As a negative control, 30 µl of 500 µM hydrogen peroxide was added to 20 µl of 0.2 mg/ml catalase (Sigma) in PBS and incubated for 10 min at room temperature, before adding to reaction solution. All samples were vortexed for 2 s and incubated at 80 °C for 7 min, before 400 µl of LC-MS extraction solution (5:3:2 methanol:acetonitrile:$H_2O$) was added. Samples were incubated in a shaker at max speed for 10 min at room temperature before centrifuging at 16,000 g for 10 min. Supernatants were transferred to glass vials and stored at −80 °C until LC-MS analysis. Phenol ($C_6H_6O$) generated by the reaction between phenylboronate and hydrogen peroxide was detected using LC-MS as described below.

**Cell proliferation assay**. Hydrogen peroxide was spiked into the medium at indicated concentrations. After 15 min, residual hydrogen peroxide was removed by aspirating and replenishing the medium. Cell proliferation was monitored by counting live cells at indicated time points.

**Colony formation assay**. Hydrogen peroxide was spiked into the medium at indicated concentrations. After 15 min, residual hydrogen peroxide was removed by aspirating the medium and cells were trypsinised. Cells were diluted 1:40 in PBS and 200 µl of cell solution was plated in six-well plates with 2 ml of medium per well. Medium was replaced every 3 days and experiments were stopped after 8 days by fixing colonies with 10% TCA overnight at 4 °C. Next day, colonies were stained by addition of 0.04% SRB in 1% acetic acid for 30 min. After four wash cycles with 1% acetic acid, plates were air dried and scanned using an Odyssey scanner (LI-COR, Inc.). Images were exported using Image Studio software (Lite Ver 5.2, LI-COR, Inc.) in grayscale (signal range 60–1800, K:1) and quantified using an ImageJ plugin designed in-house.

**Isotope tracing experiments**. Uniformly labelled glucose ($^{13}C_6$-glucose) was used at equimolar amounts as found in experimental medium (5.56 mM) and both $^{13}C_6$-glucose and 500 µM hydrogen peroxide were spiked into the medium at time point 0 and metabolites were extracted after 15 min (Fig. 3d and Fig. 5a, b). For recovery experiments (Fig. 6a–c), residual hydrogen peroxide was removed after 15 min by aspirating and replenishing the medium and metabolites were extracted at indicated time points.

**Extraction of metabolites and HPLC-MS analysis**. Metabolites were extracted and analysed as described previously[57]. Briefly, cells were washed twice with ice-cold PBS before LC-MS extraction solution (5:3:2 methanol:acetonitrile:$H_2O$) was added at 1 ml per well of a six-well plate. Plates were incubated for 5 min at 4 °C. Next, extraction solution was transferred to microcentrifuge tubes and centrifuged at 16,000g for 10 min at 4 °C. Supernatants were transferred to glass HPLC vials and stored at −80 °C until analysis LC-MS analysis. For HPLC-MS analysis a ZIC-pHILIC column (SeQuant, VWR) was used and Exactive and Q-Exactive mass spectrometers (Thermo Fisher Scientific) were operated with electrospray (ESI) ionisation and polarity switching mode at scan range ($m/z$) 75–1000 at a resolution of 25,000 at 200 $m/z$. Data were acquired using Xcalibur software (Thermo Fisher Scientific) and peak areas of metabolites were determined using TraceFinder software (Thermo Fisher Scientific) by identifying metabolites using mass and known retention time following in-house analysis of commercial standards on our systems. Metabolites were normalised to cellular protein content by Lowry assay performed on extracted cell monolayers.

**OCR measurements**. $Fh1^{fl/fl}$ cells were plated at $3 \times 10^4$ cells per well using 200 µl medium per well onto Seahorse XFe96 plates (Agilent) and incubated overnight at 37 °C and 5% $CO_2$. Then, the medium was replaced with 150 µl unbuffered Seahorse XF Base Medium (Agilent) supplemented with 5.56 mM glucose, 0.65 mM glutamine, 0.1 mM sodium pyruvate (equal to experimental medium used in other experiments) and 1% FBS (pH 7.4) and cells were placed in a $CO_2$-free incubator at 37 °C for 30 min. OCR was recorded using a Seahorse XFe96 Extracellular Flux Analyzer (Agilent) at baseline and after injection of medium (untreated condition) or indicated concentrations of hydrogen peroxide. OCR was normalised to basal OCR of untreated cells.

**Histology**. Mouse kidney tissues were fixed in 10% buffered formalin and paraffin embedded. Sections (4 µm) were stained with H&E and digital images were acquired using a Leica SCN400f slide scanner.

**Bioinformatics**. Feature Key annotations for cysteine residues were downloaded from UniProt (*Mus musculus* proteome downloaded on 03/11/2017; 51,950 sequences)[28]. Annotations from subcategories Function (Active site, Binding site, Metal binding and Site) and PTM/Processing (Disulphide bond, Modified residues and Post-translational modifications) were included and linked to Leading razor protein and Cysteine position of all identified cysteine-containing peptides using Perseus software version 1.5.5.3. The 1789 enzyme-encoding genes included in Recon 2[41] were converted to mouse associated genes using Ensembl BioMart tool (Ensembl version 85, ref. [58]), resulting in a list of 1740 mouse metabolic genes that was used for enrichment analysis. Enrichment analyses of Recon 2 and GO categories[30,31] were performed using Perseus. To further analyse the individual proteins contained in enriched GO categories, they were queried using the AmiGO web application[59]. The GOBP category of cell redox homoeostasis (GO:0045454) for *Mus musculus* was downloaded on 07/10/2017. Global protein cysteine content analysis of the metabolic and mitochondrial proteome was done based on annotation of GOBP category of metabolic process (GO:0008152) and GOCC category of mitochondrion (GO:0005739), respectively, which were downloaded for *Mus musculus* on 22/02/2017.

**Statistical methods**. The numbers of independent experiments and replicates are indicated in figure legends. Error bars represent standard deviations. For hydrogen peroxide quantification experiments, linear regression analyses were performed using GraphPad Prism 7.02 software (GraphPad Software Inc, San Diego, USA). For LFQ proteomics experiments, to determine whether protein abundance differed significantly between samples in hydrogen peroxide experiments, a two-sided $t$-test was applied using the recommended settings in Perseus software version 1.5.5.3. The eight individual LFQ measurements for untreated and hydrogen peroxide-treated samples were grouped according to treatment, no grouping was preserved in randomisations, and the number of randomisations was 250 (FDR 0.05, S0 = 0.1). To define significantly oxidised or reduced cysteine peptides for all SICyLIA proteomic experiments, the two-sided Significance B algorithm[20] was applied to the median oxidation ratio of the replicate samples for each unique peptide using Perseus. This algorithm calculates the probability that a log-ratio of at least the magnitude observed is obtained (with the Benjamini-Hochberg correction for multiple hypothesis testing). By creating intensity bins of equal occupancy, the effect of peptide intensity on their statistical spread is taken into account[60]. Cysteine peptides were considered significantly oxidised or reduced if they passed the threshold value of 0.05 (Benjamini–Hochberg FDR used for truncation). For a more detailed description of this algorithm and a substantiation of the statistical approach used, see Supplementary Note 1. Kernel density plots of the $p$-values were constructed in R version 3.4.3 using RStudio version 1.1.423 (bw = 0.4). In order to define significantly enriched GO categories within the significantly modified peptides, the Fisher exact test was applied using Perseus. Categories were considered significantly enriched if they passed the threshold value of 0.05 (Benjamini-Hochberg FDR used for truncation, with relative enrichment on leading razor protein). For Seahorse experiments, repeated measures two-way ANOVA with Dunnet's test for multiple comparisons correction was performed using GraphPad Prism 7.02 software. Significance threshold for the multiplicity adjusted $p$ value was 0.05 (95% confidence interval). No statistical method was used to predetermine sample size for the experiments.

**Reagents**. $^{13}C_6$-glucose (CLM-1396-5) was obtained from Cambridge Isotopes Laboratories (Tewksbury, MA, USA). Labelled ($^{13}C_2D_2H_2INO$, 721328) and unlabelled ($C_2H_4INO$, I6125) iodoacetamide and all other remaining reagents were obtained from Sigma-Aldrich (Merck KGaA, Darmstadt, Germany). For labelled iodoacetamide, the purity (GC), $^{13}C$ enrichment, and D enrichment were all 99% as determined by Sigma-Aldrich.

**Data availability**. The raw MS files and search/identification files obtained with MaxQuant have been deposited to the ProteomeXchange Consortium via the PRIDE partner repository[61] with the dataset identifiers PXD006363 for the $H_2O_2$ model, PXD006372 for the *Fh1* cell model, and PXD006373 for the *Fh1* tissue model. All other data from this study are available from the authors upon reasonable request.

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

## Acknowledgements

This work was supported by Cancer Research UK grants A18278 (J.vdR., L.Z., E.G.), A17196 (S.L.), A12935 and the Stand Up to Cancer campaign for Cancer Research UK (S.Z.). We would like to thank Henry Däbritz and Elaine MacKenzie for assistance with

mouse experiments, David Strachan for developing the ImageJ macro for quantification of colony formation assays, and Matthias Pietzke for assistance with R.

## Author contributions

All authors contributed to the development of the proteomics methodology. J.vdR., L.Z., and E.G. conceived the study and designed experiments. J.vdR. performed all experiments, generated the QC strategy and comparative analysis of statistical approaches, analysed and interpreted the experimental and bioinformatics data, and wrote the manuscript. S.L. processed proteomics samples, acquired and processed the data; S.L. and J.vdR. analysed and interpreted proteomics data and performed bioinformatics analyses. L.Z. developed the hydrogen peroxide quantification method. S.Z. supervised proteomics methodology development. E.G. supervised the study, interpreted data, and revised the manuscript. All authors commented on the manuscript.

## Additional information

**Competing interests:** E.G. is a Shareholder and Director at MetaboMed Ltd, Israel. The remaining authors declare no competing interests.

