## [Peer Review File · Nature Communications]

Editorial Note 1: This manuscript has been previously reviewed at another journal that is not operating a transparent peer review scheme. This document only contains reviewer comments and rebuttal letters for versions considered at Nature Communications.

Editorial Note 2: Citations to Figures in the final published Article have been amended in this Peer Review File to refer to the correct Figure.

Reviewer #1 (Remarks to the Author):

The resubmission by van der Reest and colleagues reports a strategy to determine the oxidation state of cysteine residues, relying on labeling with iodoacetamide or deuterioiodoacetamide under two conditions. NEM is also used to alkylate remaining cysteines. My initial response was that this method is indirect and requires very large amounts of mass spectrometry time. In their response the authors claimed that they wanted to create a “simple, unbiased, and robust platform to accurately quantify cysteine oxidation” on a proteome-wide scale.

My major concern was the following:

Finally, this reviewer just cannot understand how the statistics were done. There were apparently 4 separate experiments so $n=4$. However, the data were processed in bulk and compared through a “significance B” in MaxQuant. Table 1 can’t be right. For example, the first entry is Ctsd with a median ratio of 2.1 and a standard deviation of 1.8 (almost as large as the median). This peptide has a reported significance level of 2.3×10^{-142} . There are many other peptides that have questionable significance values and small reported changes with even large standard deviations. The peptide from Zadh had a median ratio of 0.4 with a standard deviation of 0.78 but a significance value of 2.1×10^{-8} . The authors have 4 replicates, they should just do a standard t-test and report the average and the std dev for those values with the p-value.

Here is the author’s response:

The referee proposes to use a t-test, however, this is not necessarily the most suitable approach to analyse these data, since every data point combines the value of the two conditions that are being compared (control vs. treatment/knock-out) into a single ratio. Therefore, the oxidation ratio of each peptide needs to be compared to all other ratios of all other detected peptides, which makes the compared groups unequal in number and not suitable for a t-test analysis. We therefore opted for the Significance B algorithm because it is an outlier significance score that highlights how significant a change in each data point is as compared to all other data points. This algorithm was specifically developed for the analysis of MS-based protein/peptide quantification using log protein/peptide ratios (<http://dx.doi.org/10.1038/nbt.1511>). This is, therefore, the most accurate statistical test for this type of data. We hope this explanation clarifies our statistical approach and we have elaborated on this more clearly in the revised manuscript and statistical methods section, as followed:

Significance B algorithm creates robust and asymmetrical estimates of the deviation from the main distribution by calculating the 15.87, 50 and 84.13 percentiles, which are right- and left-sided robust standard deviations. For normally distributed data, these are equal and meet the conventional definition of a standard deviation. These are then used to calculate a p-value to detect significant outlier ratios, defined as “Significance B”, which is the probability that a log-ratio of at least the magnitude observed is obtained (with the Benjamini-Hochberg correction for multiple hypothesis

testing, FDR <5%). Importantly, the Significance B algorithm also considers the different intensities of the peptides when calculating significant outliers, as highly abundant proteins display a more focused spread than low abundant ones (which reflects the accuracy of the calculated peptide ratios: more intense peptides are generally quantified based on a higher number of quantified events). It does this by creating intensity groups (bins) of equal occupancy for all peptides, over which the significance is calculated.

To address the other concern: The referee mentioned significance and standard deviation values reported in the data tables. We want to highlight that there is no relation between the standard deviation and the significance B value reported, as the “Significance B” is calculated only on the median of the 4 replicate experiments and does not consider the standard deviation between replicates. For this reason, we included the standard deviation in the tables, to provide readers insight into the variability. Unfortunately this has led to a misinterpretation. We want to reassure this referee that overall variability in our datasets is low. The median coefficient of variation (CV%) between the 4 replicates for the quantified cysteine-containing peptides was 16.0 for hydrogen peroxide-treated cells, 18.6 for the FH cell model, and 17.3 for the FH tissue model. This compares favourably to another recent quantitative analysis of peptide cysteine oxidation induced by hydrogen peroxide treatment that used enrichment steps, where the median CV was 26 (<http://dx.doi.org/10.1038/ncomms5776>), and the CV% was 20 in the in-depth proteome analysis using high pH fractionation that we mentioned before (<http://dx.doi.org/10.1016/j.cels.2017.05.009>). For this reason, we felt it was justified to use the median of our 4 independent experiments for statistical analysis. However, as this referee correctly points out, there are indeed some outliers with high variability, and for this reason we have reported the standard deviation of the median in the tables.

However, to better take into account the variability between the 4 replicates and address the variability concerns of this reviewer, we have included additional stringent quality control measures to achieve accurate cysteine oxidation ratios. First, only peptide oxidation ratios that were robustly quantified in at least three replicates were considered for analysis. Additionally, the coefficient of variation between replicates was used to filter out extreme outlier ratios. This was done by calculating upper fences ($Q3 + 1.5IQR$, where $Q3$ is the third quartile and IQR is the interquartile range) of the distribution of CVs for all median peptide oxidation ratios in each dataset and using these as cut-off. This led to the exclusion of 533 (H₂O₂ model), 491 (Fh1 cell model), and 310 (Fh1 tissue model) peptides. After quality control, the median peptide and protein ratios between replicates were calculated and used for all further analyses to minimise the effect of outliers. After QC, variability across our datasets was further reduced and replicates showed high reproducibility, with median CVs of 15.3 (H₂O₂ model), 17.6 (Fh1 cell model), and 16.2 (Fh1 tissue model). We believe that these additional quality control steps have made our analysis stronger and we hope this addresses the concerns raised by the referee.

We have included detailed description of quality control steps and statistical analyses in the manuscript and relevant methods sections. We also included a figure that highlights the high reproducibility and performance indicators of the SiCyLIA workflow (Figure 2). Finally, separate figures are included for all proteomics experiments that use the median between replicates to show the standard deviation between individual replicate experiments (Supplementary Fig. 2c, Supplementary Fig. 3a, Supplementary Fig. 4a). The subsequent enrichment analyses and other results and conclusions are not significantly affected by this modified analysis.

After carefully reading the author's response, I remain unconvinced that this is the correct way to analyze these data. The heart of the issue here is that the authors are ignoring the actual experimental variability and make no use of the biological replicates to assess variation. The statistic used by the authors only asks where in the distribution of all value ranges does the value fall. Several things could be done. The easiest might be a 1-sample t-test and setting the null hypothesis to be that the Log₂ of the ratio is 0. Alternatively, don't take the ratio and do a paired t-test with the AUC of each sample. Finally, a linear model could incorporate even more information. The key here is that the experimental error is known, but not used in assessing significance. I cannot support this as a proper way to do this analysis. The "significance B" idea came from not having replicates. In a SILAC experiment, there was just n=1 for comparing two states. They needed a way to determine the significance of ratios. This approach is flawed for the very reasons that were pointed out. Clearly a peptide with a mean ratio of 2.1 and a standard deviation of 1.8 will not be significant in any test that uses the experimental variability. I stand by my previous comments. Appropriate statistics were not used.

Reviewer #3 (Remarks to the Author):

The authors have responded to all previous reviewer comments adequately. I therefore recommend publication of this manuscript in its current form.

The authors would like to thank the Referees for assessing our manuscript “Proteome-wide analysis of cysteine oxidation reveals metabolic sensitivity to redox stress”. Their interest and time are greatly appreciated. In response to our revised manuscript, Reviewer#1 continues to raise concerns regarding the statistical analysis of the data and proposed an alternative strategy that depends on t-test analysis. We have followed the Referee’s suggestions and re-analyzed our data, and outline our findings and response to the Referee below in green

Reviewer’s comments:

Reviewer #1 (Remarks to the Author):

The resubmission by van der Reest and colleagues reports a strategy to determine the oxidation state of cysteine residues, relying on labeling with iodoacetamide or deuterioiodoacetamide under two conditions. NEM is also used to alkylate remaining cysteines. My initial response was that this method is indirect and requires very large amounts of mass spectrometry time. In their response the authors claimed that they wanted to create a “simple, unbiased, and robust platform to accurately quantify cysteine oxidation” on a proteome-wide scale.

My major concern was the following:

Finally, this reviewer just cannot understand how the statistics were done. There were apparently 4 separate experiments so $n=4$. However, the data were processed in bulk and compared through a “significance B” in MaxQuant. Table 1 can’t be right. For example, the first entry is Ctsd with a median ratio of 2.1 and a standard deviation of 1.8 (almost as large as the median). This peptide has a reported significance level of 2.3×10^{-142} . There are many other peptides that have questionable significance values and small reported changes with even large standard deviations. The peptide from Zadh had a median ratio of 0.4 with a standard deviation of 0.78 but a significance value of 2.1×10^{-8} . The authors have 4 replicates, they should just do a standard t-test and report the average and the std dev for those values with the p-value.

Here is the author’s response:

The referee proposes to use a t-test, however, this is not necessarily the most suitable approach to analyse these data, since every data point combines the value of the two conditions that are being compared (control vs. treatment/knock-out) into a single ratio. Therefore, the oxidation ratio of each peptide needs to be compared to all other ratios of all other detected peptides, which makes the compared groups unequal in number and not suitable for a t-test analysis. We therefore opted for the Significance B algorithm because it is an outlier significance score that highlights how significant a change in each data point is as compared to all other data points. This algorithm was specifically developed for the analysis of MS-based protein/peptide quantification using log protein/peptide ratios (<http://dx.doi.org/10.1038/nbt.1511>). This is, therefore, the most accurate statistical test for this type of data. We hope this explanation clarifies our statistical approach and we have elaborated on this more clearly in the revised manuscript and statistical methods section, as followed:

Significance B algorithm creates robust and asymmetrical estimates of the deviation from the main distribution by calculating the 15.87, 50 and 84.13 percentiles, which are right- and left-sided robust standard deviations. For normally distributed data, these are equal and meet the conventional definition of a standard deviation. These are then used to calculate a p-value to detect significant outlier ratios, defined as “Significance B”, which is the probability that a log-ratio of at least the magnitude observed is obtained (with the Benjamini-Hochberg correction for multiple hypothesis testing, FDR <5%). Importantly, the Significance B algorithm also considers the different intensities of the peptides when calculating significant outliers, as highly abundant proteins display a more focused spread than low abundant ones (which reflects the accuracy of the calculated peptide ratios: more intense peptides are generally quantified based on a higher number of quantified events). It does this by creating intensity groups (bins) of equal occupancy for all peptides, over which the significance is calculated.

To address the other concern: The referee mentioned significance and standard deviation values reported in the data tables. We want to highlight that there is no relation between the standard deviation and the significance B value reported, as the “Significance B” is calculated only on the median of the 4 replicate experiments and does not consider the standard deviation between replicates. For this reason, we included the standard deviation in the tables, to provide readers insight into the variability. Unfortunately this has led to a misinterpretation. We want to reassure this referee that overall variability in our datasets is low. The median coefficient of variation (CV%) between the 4 replicates for the quantified cysteine-containing peptides was 15.9 for hydrogen peroxide-treated cells, 18.6 for the FH cell model, and 17.1 for the FH tissue model. This compares favourably to another recent quantitative analysis of peptide cysteine oxidation induced by hydrogen peroxide treatment that used enrichment steps, where the median CV was 26 (<http://dx.doi.org/10.1038/ncomms5776>), and the CV% was 20 in the in-depth proteome analysis using high pH fractionation that we mentioned before (<http://dx.doi.org/10.1016/j.cels.2017.05.009>). For this reason, we felt it was justified to use the median of our 4 independent experiments for statistical analysis. However, as this referee correctly points out, there are indeed some outliers with high variability, and for this reason we have reported the standard deviation of the median in the tables.

However, to better take into account the variability between the 4 replicates and address the variability concerns of this reviewer, we have included additional stringent quality control measures to achieve accurate cysteine oxidation ratios. First, only peptide oxidation ratios that were robustly quantified in at least three replicates were considered for analysis. Additionally, the coefficient of variation between replicates was used to filter out extreme outlier ratios. This was done by calculating upper fences ($Q3 + 1.5IQR$, where $Q3$ is the third quartile and IQR is the interquartile range) of the distribution of CVs for all median peptide oxidation ratios in each dataset and using these as cut-off. This led to the exclusion of 579 (H_2O_2 model), 499 (Fh1 cell model), and 315 (Fh1 tissue model) peptides. After quality control, the median peptide and protein ratios between replicates were calculated and used for all further analyses to minimise the effect of outliers. After QC, variability across our datasets was further reduced and replicates showed high reproducibility, with median CVs of 15.1 (H_2O_2 model), 17.7 (Fh1 cell model), and 16.0 (Fh1 tissue model). We believe that these additional quality control steps have made our analysis stronger and we hope this addresses the concerns raised by the referee.

We have included detailed description of quality control steps and statistical analyses in the manuscript and relevant methods sections. We also included a figure that highlights the high reproducibility and performance indicators of the SiCyLIA workflow (Figure 2). Finally, separate figures are included for all proteomics experiments that use the median between replicates to show the standard deviation between individual replicate experiments (Supplementary Fig. 2c, Supplementary Fig. 3a, Supplementary Fig. 4a). The subsequent enrichment analyses and other results and conclusions are not significantly affected by this modified analysis.

After carefully reading the author's response, I remain unconvinced that this is the correct way to analyze these data. The heart of the issue here is that the authors are ignoring the actually experimental variability and make no use of the biological replicates to assess variation. The statistic used by the authors only asks where in the distribution of all value ranges does the value fall. Several things could be done. The easiest might be a 1-sample t-test and setting the null hypothesis to be that the Log₂ of the ratio is 0. Alternatively, don't take the ratio and do a paired t-test with the AUC of each sample. Finally, a linear model could incorporate even more information. The key here is that the experimental error is known, but not used in assessing significance. I cannot support this as a proper way to do this analysis. The "significance B" idea came from not having replicates. In a SILAC experiment, there was just n=1 for comparing two states. They needed a way to determine the significance of ratios. This approach is flawed for the very reasons that were pointed out. Clearly a peptide with a mean ratio of 2.1 and a standard deviation of 1.8 will not be significant in any test that uses the experimental variability. I stand by my previous comments. Appropriate statistics were not used.

Authors' response:

In a nutshell, the criticism of the Referee concerns our statistical approach ("Significance B" algorithm vs. t-test). The Referee states that "the authors are ignoring the actual experimental variability between replicates and make no use of the biological replicates to assess variation". However, we have in fact used the biological replicates and the variation between them to filter out peptides which are statistical outliers based on their coefficient of variation (CV%) in comparison to CV% of all identified peptides. We took this approach in acknowledging the Referee's original comments on our first submission of the manuscript, and this was explained in our previous rebuttal letter and our latest submission to Nature Communications. Unfortunately, Referee 1 did not seem to recognise this important modification we included in the revised manuscript (in response to his/her original request), and repeated his/her original argument on variable peptides, which in fact are no longer part of the study due to the filtering steps we have introduced. More specifically, we would like to point out that all information regarding variability for every single peptide identified is available for readers, as we provided the CV% and the standard deviation (SD) in the data tables, and the Supplementary Information contains figures displaying the oxidation ratio versus the SD for each peptide quantified. Additionally, only peptides identified in at least 3 replicate experiments are included, and for all analyses the median of the replicate experiments is used to further minimise the influence of variation. Because statistical outliers are filtered out as a quality control step based on their CV%, the example that the Referee gave (ratio of 2.1 with SD of 1.8) from the original manuscript does not pass quality control and is therefore filtered out after introducing the CV% filtering step. As such, our analysis strategy is very much designed to take into account replicates and variability. We would like to highlight that biological cysteine oxidation is a transient event that

is rapidly turned over. As such, peptide oxidation measurements are inherently variable. Therefore, an appropriate statistical test needs to be able to identify biologically-relevant redox-sensitive peptides despite substantial variability in peptide oxidation measures. This consideration was central to the design of our analysis strategy. Nevertheless, we have carefully considered these important points and tested multiple methods for statistical analyses of our data which are now discussed below and included in a Supplementary Note in the manuscript.

The Referee highlights that the “Significance B” algorithm that we use to assess significance was developed for single-replicate experiments. As we use four replicate experiments, he/she proposed two types of t-test analyses that will take into account these replicates. One suggestion was the paired t-test that would utilize all four replicate intensity values (AUC) of each peptide under each condition (control or oxidative stress). However, such a strategy is counterintuitive to the multiplexing approach that we purposefully included in our methodology, where replicate samples are mixed and ratios are normalised in MaxQuant at once, equal to in SILAC experiments. We chose this approach in order to decrease technical variabilities introduced by sample processing and, most importantly, to specifically focus on the **ratio of change** (oxidised/reduced), as this is the biologically meaningful readout that this methodology was designed to obtain. Using the single intensity values instead of ratios would require normalisation steps of the separate intensities, which could result in reduced accuracy of the quantification compared to what is achieved in MaxQuant when using the normalised ratios between labelled peptide pairs.

The other proposed analysis was the 1-sample t-test under the null hypothesis that the log₂ oxidation ratio is 0 (with 0 meaning no change in oxidation state). We applied this test to our data: shown below are the results for the dataset of hydrogen peroxide-treated cells.

Both figures show scatterplots of oxidation ratios of all peptides included in the analysis, plotted against their intensity. Significant hits are highlighted in green. The left panel shows our original “Significance B” analysis (333 hits), the right panel shows the t-test results ($p < 0.05$) as proposed by the Referee (327 hits). As one can appreciate, the peptides with oxidation ratios that change most dramatically after treatment (both positively and negatively) are considered significant using both

analyses. In total, 117 hits pass significance using both tests. However, you can also appreciate that the t-test favours many peptides with small ratio values, which are found in the middle of the distribution. These peptides undergo only a marginal change in oxidation state, which is exemplified by an oxidation ratio close to 0, but have low experimental variability and therefore pass significance using the t-test. We would like to stress that our methodology aims to identify redox-reactive peptides, and in fact, many peptides that undergo larger changes in oxidation state (and are thus most interesting to identify from a biological perspective) do not pass significance using the t-test because their oxidation ratios are inherently more variable. This is more clearly demonstrated in the following figures:

Again, both figures show scatterplots of oxidation ratios of all peptides included in the analysis, plotted against their intensity. Peptides are now coloured by the value of the CV% between the replicate experiments, scale indicated above. Significant hits using Significance B (left) and the t-test (right) are highlighted by large filled circle symbols. As you can appreciate, the CV% increases for peptides with larger oxidation ratios and also those of lower intensity (blue colours). Clearly, the t-test favours those peptides with low CV% (green, yellow, orange colours). However, a large proportion of these are also the peptides with small changes in oxidation ratio, which from a biological relevance perspective are not the hits we aim to identify with this methodology. Additionally, the t-test uses the same stringency for peptides of low and higher intensity, whereas the low intensity peptides are inherently more variable than highly abundant ones (<http://dx.doi.org/10.1038/nbt.1511>, <http://dx.doi.org/10.1007/s00216-007-1486-6>). The Significance B algorithm is, to our knowledge, the only test that takes protein/peptide intensity into account by creating intensity groups (bins).

This is further exemplified by the plot below, which shows the oxidation ratio of all peptides identified plotted against their CV%. Highlighted in green are the hits that pass significance using t-test analysis.

This plot clearly shows that variation is the main parameter that determines whether a peptide will be considered significant using the t-test, despite many of them having an oxidation ratio close to 0 and thus marginal changes in oxidation state (highlighted in the figure above). As the aim of the methodology is to identify peptides that are meaningfully redox-reactive, and thus those with a larger positive or negative oxidation ratio, this indicates that the t-test is not suitable to the aim of the methodology. **In contrast, the significance B algorithm selects those peptides that have undergone a significantly larger change in oxidation state compared to all other peptides of similar intensity identified, and thus, identifies the most redox-sensitive peptides in the dataset.**

Equal to this referee, we also fully appreciate and support the importance of reproducibility of results and stringency of analyses. The above analysis shows that variation in the quantified peptide oxidation ratios is inherent for peptides that undergo the largest biological change in oxidation, or those that are of lower abundance. However, we found that this increased variation does not reflect a lack of biological significance. We would like to illustrate this by highlighting some top hits identified using our methodology. These are bona fide redox targets of known function, which have been extensively shown to play a role in cellular redox biology by researchers in the field (GAPDH, PARK7, PTEN, Thioredoxin 2, and the Peroxiredoxin family members):

Gene name	CV%	Oxidation ratio	Cysteine biological function
Gapdh	32	2.4	Active site
Park7	21	1.0	Active site
Pten	18	2.7	Active site
Txn2	44	2.1	Disulphide bond
Prdx1	49	-1.8	Disulphide bond
Prdx2	47	-2.8	Disulphide bond
Prdx2	43	-2.7	Disulphide bond
Prdx4	33	-1.1	Disulphide bond
Prdx5	21	-1.0	Disulphide bond
Prdx6	16	-0.5	Active site

There is a wide range of variabilities for these peptides, with all showing higher CV% as compared to the median CV% of all identified peptides (which is 15.1 in this dataset). All of these peptides pass significance using t-test analysis too, driven by their large change in oxidation ratio. However, in addition to these cysteine peptides known to be extremely reactive (peroxiredoxins are the most reactive redox proteins in the cell), we are also interested in identifying peptides that have milder changes in oxidation states, because they can also be biologically relevant redox-sensitive proteins. This is a key aim of our methodology: due to its unprecedented depth compared to previous methods, it can be used as a screening tool to identify novel redox-sensitive proteins besides the highly reactive redox sensors that are already well-characterised. The more mildly reactive peptides are those that can be seen on the edges of the data distribution, as their reactivity stands out from the total amount of peptides in the dataset, of which the majority (>96%) is not redox-regulated. These peptides are identified using the Significance B algorithm. However, due to the inherent variability of biological cysteine oxidation they do not pass the t-test. Therefore, this test does not fit the aim of this methodology: we are interested in highlighting those peptides that have larger oxidation ratios compared to other peptides, as they indicate a stronger biological effect in the cell. This information is missed when relying solely on the t-test.

We would like to stress again that the oxidation ratios are measured at the peptide level. Such measurements are inherently more variable than measurements at the *protein* level, which are based on measurements of tens of individual peptides. However, that variation at the peptide oxidation level does not indicate poor biological reproducibility. This is illustrated by the comparison of a miscleaved peptide of Peroxiredoxin 2:

Gene name	CV%	Oxidation ratio	Sequence
Prdx2	47	-2.78	LVQAFQYTDEHGEVCPAGWK
Prdx2 (miscleavage)	43	-2.73	LVQAFQYTDEHGEVCPAGWKPGSDTIK

Due to the miscleavage, these peptides are identified as two unique peptides and their oxidation ratios are quantified separately. But biologically, they are derived from the exact same peptide/protein population. As you can appreciate, the oxidation ratio for both is nearly identical, despite having a relatively high CV% of over 40. This clearly shows that a high variation, indicated by high CV%, does not necessarily mean high biological variability, as the resulting peptide oxidation ratio is highly reproducible.

We believe we have taken many steps to ensure reproducible results: peptides need to be robustly quantified in at least three replicate experiments, outliers are filtered out, the median is used throughout, and the CV% and SD for each peptide are provided for readers. Additionally, the Significance B algorithm is a stringent analysis due to the use of the Benjamini-Hochberg FDR, and uses intensity binning to account for peptide intensity (and thus variability) differences. As such, we are convinced that our approach has sufficiently taken experimental variability into account. As detecting peptide cysteine oxidation on whole proteome scale is a matter of locating small differences within big background noise, we believe, and showed in the above analyses, that **the application of multi-step filtering rather than relying solely on a t-test to select results with**

biological relevance is well-suited to the type of datasets generated with the SiCyLIA methodology.

Finally, we have analysed all datasets using our Significance B approach and then further applied the 1-sample t-test that the Referee suggested, and this showed that the number of hits that passed both tests is 117 (H₂O₂ model), 123 (*Fh1* cell model), and 30 (*Fh1* kidney model). We have repeated the enrichment analyses using only these hits that passed both statistical tests, and it achieved the same global results obtained using Significance B analysis alone, thus further supporting the validity of using the Significance B approach. Despite having fewer proteins in all enriched categories due to the decreased number of peptides that are considered significant using this approach, because only highly redox-reactive proteins are included, there is still significant enrichment of mitochondrial and metabolic proteins in the acute oxidative stress model; and enrichment of metabolic proteins in the chronic stress condition. Moreover, we still see that the metabolic proteins oxidised in acute and chronic oxidative stress are distinct, whereas the significantly enriched dedicated redox proteins show greater overlap.

Despite this, we are convinced that: 1) the t-test leads to inclusion of many peptides that are well-identified but do not actually undergo a meaningful redox change, and 2) fails to identify certain biologically relevant redox-sensitive proteins. For the first reason, we believe that using solely the t-test for statistical analysis on these data does not support the aim of the methodology. For the second reason, we believe that using the combination of both tests and including only hits that pass both tests, despite leading to the same global results, is not useful to identify individual mildly redox-sensitive proteins that will be interesting and meaningful for fellow redox biologists. An important aim of publishing the global proteomic datasets of unique depth that we have generated is to provide a resource of newly identified redox-sensitive proteins that can be further characterised in follow-up studies. For this reason, we strongly support the use of our original Significance B analysis.

After editorial consultation, we have decided to include a modified version of the above analysis in our revised manuscript as a Supplementary Note to provide insight into the rationale underlying our multi-step filtering approach and statistical analysis. We believe that this analysis strengthens our manuscript by rationalising the Significance B approach as compared to t-test analysis, while further elaborating on the strengths and weaknesses of both tests for peptide oxidation analysis. We also extensively discuss the variability of peptide oxidation measures, and stress that an important consideration of using Significance B analysis is that biological replicates are not fully considered when assessing significance. Therefore, the CV% and standard deviation between replicate experiments of each peptide are provided in the manuscript to aid the selection of the most meaningful hits. We hope this added analysis will be of interest to the proteomics community readership, and will allow fellow researchers to select the analysis strategy best suited to the aim of their particular experiments when using SiCyLIA. We also hope that this shows that we have taken the Referee's concerns to heart in the design of our analysis strategy, and have clearly communicated the important considerations raised by the Referee to readers.

Thank you very much for your time and assistance. On behalf of all authors, with best wishes,

Eyal Gottlieb

Reviewer #4:

Remarks to the Author:

After careful digestion of the content of the manuscript and the communications between the authors and Reviewer 1, I would like to summarize the key differences, express my opinions and suggest some minor modifications to the authors.

1) Significance B test vs one-sample t-test

Significance B test is inherently a one-sample test, similar to 1-sample t-test. As the authors pointed out in a reply, the key advantage of Significance B test is it takes the intensities of the peptides into account as well for the estimation of statistical significance, while 1-sample t-test does not, therefore a better test. That is indeed the case as knowledge in the field and figure-illustrated by the authors in the communication. Therefore, if one accepts the logic that median ratio values from replicates should be taken to construct a virtual one-sample, Significance B test would be serve as a better statistical method in this scenario.

2) paired t-test or not

Reviewer 1 also suggested calculating the intensity difference and significance by paired t-test (volcano-plot like approach/representation). However, as authors pointed out, the whole experimental design of the SICyLIA methodology to label and mix samples at two states renders a purpose to report intensity ratios by minimizing the variability generated from sample processing. Therefore, on this issue, I support authors' approach.

3) Combine replicates or not

The authors took median ratio values to construct a virtual one-sample and reported standard deviation for data variability. This is not invalid but still certain amount of information has been lost. To increase transparency, I think ratio values from all replicates should be reported in the Supplementary Table, which is only a matter of adding a few columns.

4) Assessment of the robustness of statistical analysis

As pointed out, the authors actually have obtained ratio values of all the replicates but constructed a single experiment by taking the medians. This is presumably for simplicity of analysis and representation but certainly render a loss of information for the assessment the robustness of statistical performance. I suggest in addition to the median ratio value approach, the authors conduct Significance B test for each replicate and report the p values individually, to further increase

the transparency and reveal the degree of robustness. Calculation of MAD or Std of the p-values of each replicate and generating a distribution plot such like kernel-density plot would be an ideal way to represent the robustness. This analysis is simple and again, adding a few columns of p values in the Supplementary Table will increase the transparency of the computational pipeline.

If robustness is sufficient, I support publication of the manuscript.

The authors would like to thank Reviewer #4 for assessing our manuscript “Proteome-wide analysis of cysteine oxidation reveals metabolic sensitivity to redox stress” and our communications with Reviewer #1. The Reviewer’s time and helpful suggestions are greatly appreciated. We outline the comments and our response below (in green).

Reviewer’s comments:

Reviewer #4 (Remarks to the Author):

After careful digestion of the content of the manuscript and the communications between the authors and Reviewer 1, I would like to summarize the key differences, express my opinions and suggest some minor modifications to the authors.

1) Significance B test vs one-sample t-test

Significance B test is inherently a one-sample test, similar to 1-sample t-test. As the authors pointed out in a reply, the key advantage of Significance B test is it takes the intensities of the peptides into account as well for the estimation of statistical significance, while 1-sample t-test does not, therefore a better test. That is indeed the case as knowledge in the field and figure-illustrated by the authors in the communication. Therefore, if one accepts the logic that median ratio values from replicates should be taken to construct a virtual one-sample, Significance B test would be serve as a better statistical method in this scenario.

Thank you.

2) paired t-test or not

Reviewer 1 also suggested calculating the intensity difference and significance by paired t-test (volcano-plot like approach/representation). However, as authors pointed out, the whole experimental design of the SICyLIA methodology to label and mix samples at two states renders a purpose to report intensity ratios by minimizing the variability generated from sample processing. Therefore, on this issue, I support authors’ approach.

Thank you.

3) Combine replicates or not

The authors took median ratio values to construct a virtual one-sample and reported standard deviation for data variability. This is not invalid but still certain amount of information has been lost. To increase transparency, I think ratio values from all replicates should be reported in the Supplementary Table, which is only a matter of adding a few columns.

The Reviewer is correct that median ratio values are used to construct a virtual one-sample. To ensure that information regarding the variability within the four replicates is known to readers, we have reported the standard deviation and coefficient of variation (CV%) between replicates in the data tables. The Reviewer suggests that also reporting the individual ratio values for each replicate further increases transparency, which we totally agree with, and in fact this was already reported in

the data tables. In addition to the log₂ Median Peptide Oxidation Ratio we also included the log₂ Peptide Oxidation Ratio of Replicate 1/2/3/4 for each replicate.

To ensure that readers are aware of the different values reported in the data tables and can locate information contained therein more easily, we have added a tab with “Definitions” to each data table where we define each data category (column headers). We hope this will make the data tables more understandable and accessible.

4) Assessment of the robustness of statistical analysis

As pointed out, the authors actually have obtained ratio values of all the replicates but constructed a single experiment by taking the medians. This is presumably for simplicity of analysis and representation but certainly render a loss of information for the assessment the robustness of statistical performance. I suggest in addition to the median ratio value approach, the authors conduct Significance B test for each replicate and report the p values individually, to further increase the transparency and reveal the degree of robustness. Calculation of MAD or Std of the p-values of each replicate and generating a distribution plot such like kernel-density plot would be an ideal way to represent the robustness. This analysis is simple and again, adding a few columns of p values in the Supplementary Table will increase the transparency of the computational pipeline.

If robustness is sufficient, I support publication of the manuscript.

We have followed the suggestions made by the Reviewer to further increase transparency and reveal the degree of robustness of our study. We performed Significance B analysis on the individual replicates in addition to the analysis of the median ratio values. The resulting p-values for each replicate are now added to the data tables (“Significance B value Peptide Oxidation Ratio Replicate 1/2/3/4”).

Peptides with the largest median oxidation ratios, which thus changed their oxidation states most dramatically, generally pass significance on the individual replicate level as well. However, peptides on the edges of the data distribution may not pass significance in each individual replicate. These peptides show milder changes in oxidation state that may nevertheless be biologically relevant, but as such may not be among the top hits in each individual replicate. This relates to the inherent variability that we have outlined in the Supplementary Note and previous communication.

Additionally, there are peptides in the dataset that are more reduced (negative value) in one replicate, but more oxidised (positive value) in another. This may reflect a true biological effect: cysteine residues can inherently exist in a reduced or oxidised state. These peptides may pass significance in several individual replicates, but their median oxidation ratio will be close to 0 and thus not considered significant. As such, using the median oxidation ratio for analysis is a choice we purposefully made: we aimed to identify with Significance B those peptides that consistently showed oxidation or reduction, and thus have a median negative or positive oxidation ratio. This does not mean, however, that the peptides that are reduced in some replicates and oxidised in others are not biologically meaningful redox-sensitive peptides; they might be, but we cannot conclude this from having essentially only two observations where they are either reduced or oxidised. Therefore, we have filtered out peptides with a large CV% and have used the median ratio to limit our analyses to those peptides that consistently change their oxidation state in the same direction.

These are important considerations when comparing the median Significance B value and the individual Significance B values for each replicate. We have included these considerations in the Supplementary Note, as they explain that the median and individual Significance B values are not necessarily expected to be the same and so the readers can assess the choices underlying our analysis strategy.

Next, the Reviewer says: “Calculation of MAD or Std of the p-values of each replicate and generating a distribution plot such like kernel-density plot would be an ideal way to represent the robustness”.

Below, we show kernel density plots of the distribution of p-values for each replicate experiment of the three models. The vertical grey line indicates the p-value of 0.05 used as threshold for significance in our study; we also report the mean and standard deviation (SD) of the distributions:

Fh1 cell model

Fh1 tissue model

As can be appreciated, the kernel density plots and standard deviations of the p-values show a high degree of similarity for the four replicates of each experimental model. Thus, we believe these additions reveal the robustness of the statistical analysis and further substantiate our median ratio value approach. As the Reviewer raised important issues regarding the robustness of statistical analysis, which should be considered in future applications of the methodology, we have included the above analyses and considerations in the Supplementary Note.

We hope we have satisfactorily addressed the requests of the Reviewer in this letter, and once again want to thank the Reviewer and editorial panel for their time and helpful suggestions to improve our manuscript.